# Stochastic Approximation
# for Canonical Correlation Analysis

**Raman Arora**
Dept. of Computer Science
Johns Hopkins University
Baltimore, MD 21204
arora@cs.jhu.edu

**Teodor V. Marinov**
Dept. of Computer Science
Johns Hopkins University
Baltimore, MD 21204
tmarino2@jhu.edu

**Poorya Mianjy**
Dept. of Computer Science
Johns Hopkins University
Baltimore, MD 21204
mianjy@jhu.edu

**Nathan Srebro**
TTI-Chicago
Chicago, Illinois 60637
nati@ttic.edu

## Abstract

We propose novel first-order stochastic approximation algorithms for canonical correlation analysis (CCA). Algorithms presented are instances of inexact matrix stochastic gradient (MSG) and inexact matrix exponentiated gradient (MEG), and achieve $\epsilon$-suboptimality in the population objective in $\text{poly}(\frac{1}{\epsilon})$ iterations. We also consider practical variants of the proposed algorithms and compare them with other methods for CCA both theoretically and empirically.

## 1 Introduction

Canonical Correlation Analysis (CCA) [11] is a ubiquitous statistical technique for finding maximally correlated linear components of two sets of random variables. CCA can be posed as the following stochastic optimization problem: given a pair of random vectors $(x, y) \in \mathbb{R}^{d_x} \times \mathbb{R}^{d_y}$, with some (unknown) joint distribution $\mathscr{D}$, find the $k$-dimensional subspaces where the projections of $x$ and $y$ are maximally correlated, i.e. find matrices $\tilde{U} \in \mathbb{R}^{d_x \times k}$ and $\tilde{V} \in \mathbb{R}^{d_y \times k}$ that

$$\text{maximize } \mathbb{E}_{x,y}[x^\top \tilde{U}\tilde{V}^\top y] \text{ subject to } \tilde{U}^\top \mathbb{E}_x[xx^\top]\tilde{U} = I_k, \tilde{V}^\top \mathbb{E}_y[yy^\top]\tilde{V} = I_k. \tag{1}$$

CCA-based techniques have recently met with success at unsupervised representation learning where multiple "views" of data are used to learn improved representations for each of the views [3, 5, 13, 23]. The different views often contain complementary information, and CCA-based "multiview" representation learning methods can take advantage of this information to learn features that are useful for understanding the structure of the data and that are beneficial for downstream tasks.

Unsupervised learning techniques leverage unlabeled data which is often plentiful. Accordingly, in this paper, we are interested in first-order stochastic Approximation (SA) algorithms for solving Problem (1) that can easily scale to very large datasets. A stochastic approximation algorithm is an iterative algorithm, where in each iteration a single sample from the population is used to perform an update, as in stochastic gradient descent (SGD), the classic SA algorithm.

There are several computational challenges associated with solving Problem (1). A first challenge stems from the fact that Problem (1) is non-convex. Nevertheless, akin to related spectral methods such as principal component analysis (PCA), the solution to CCA can be given in terms of a generalized eigenvalue problem. In other words, despite being non-convex, CCA admits a tractable

algorithm. In particular, numerical techniques based on power iteration method and its variants can be applied to these problems to find globally optimal solutions. Much recent work, therefore, has focused on analyzing optimization error for power iteration method for the generalized eigenvalue problem [1, 8, 24]. However, these analyses are on *numerical (empirical) optimization error* for finding left and right singular vectors of a fixed given matrix based on empirical estimates of the covariance matrices, and not on the population $\epsilon-$suboptimality (aka bound in terms of population objective) of Problem (1) which is the focus here.

The second challenge, which is our main concern here, presents when designing first order stochastic approximation algorithms for CCA. The main difficulty here, compared to PCA, and most other machine learning problems, is that the constraints also involve stochastic quantities that depend on the unknown distribution $\mathscr{D}$. Put differently, the CCA objective does not decompose over samples. To see this, consider the case for $k = 1$. The CCA problem then can be posed equivalently as maximizing the correlation objective $\rho(\mathrm{u}^T\mathrm{x}, \mathrm{v}^T\mathrm{y}) = \mathbb{E}_{\mathrm{x},\mathrm{y}} \left[\mathrm{u}^\top \mathrm{xy}^\top \mathrm{v}\right] / (\sqrt{\mathbb{E}_\mathrm{x}\left[\mathrm{u}^\top \mathrm{xx}^\top \mathrm{u}\right]} \sqrt{\mathbb{E}_\mathrm{y}\left[\mathrm{v}^\top \mathrm{yy}^\top \mathrm{v}\right]})$. This yields an unconstrained optimization problem. However, the objective is no longer an expectation, but is instead a ratio of expectations. If we were to solve the empirical version of this problem, it is easy to check that the objective ties all the samples together. This departs significantly from typical stochastic approximation scenario. Crucially, with a single sample, it is not possible to get an unbiased estimate of the gradient of the objective $\rho(\mathrm{u}^T\mathrm{x}, \mathrm{v}^T\mathrm{y})$. Therefore, we consider a first-order oracle that provides inexact estimates of the gradient with a norm bound on the additive noise, and focus on inexact proximal gradient descent algorithms for CCA.

Finally, it can be shown that the CCA problem given in Problem (1) is ill-posed if the population auto-covariance matrices $\mathbb{E}_\mathrm{x}\left[\mathrm{xx}^\top\right]$ or $\mathbb{E}_\mathrm{y}\left[\mathrm{yy}^\top\right]$ are ill-conditioned. This observation follows from the fact that if there exists a direction in the kernel of $\mathbb{E}_\mathrm{x}\left[\mathrm{xx}^\top\right]$ or $\mathbb{E}_\mathrm{y}\left[\mathrm{yy}^\top\right]$ in which x and y exhibit non-zero covariance, then the objective of Problem (1) is unbounded. We would like to avoid recovering such directions of spurious correlation and therefore assume that the smallest eigenvalues of the auto-covariance matrices and their empirical estimates are bounded below by some positive constant. Formally, we assume that $\mathrm{C}_x \succeq r_x\mathrm{I}$ and $\mathrm{C}_y \succeq r_y\mathrm{I}$. This is the typical assumption made in analyzing CCA [1, 7, 8].

## 1.1 Notation

Scalars, vectors and matrices are represented by normal, Roman and capital Roman letters respectively, e.g. $x$, x, and X. $\mathrm{I}_k$ denotes identity matrix of size $k \times k$, where we drop the subscript whenever the size is clear from the context. The $\ell_2$-norm of a vector x is denoted by $\|\mathrm{x}\|$. For any matrix X, spectral norm, nuclear norm, and Frobenius norm are represented by $\|\mathrm{X}\|_2$, $\|\mathrm{X}\|_*$, and $\|\mathrm{X}\|_F$ respectively. The trace of a square matrix X is denoted by $\mathrm{Tr}\,(\mathrm{X})$. Given two matrices $\mathrm{X} \in \mathbb{R}^{k \times d}$, $\mathrm{Y} \in \mathbb{R}^{k \times d}$, the standard inner-product between the two is given as $\langle \mathrm{X}, \mathrm{Y}\rangle = \mathrm{Tr}\,(\mathrm{X}^\top \mathrm{Y})$; we use the two notations interchangeably. For symmetric matrices X and Y, we say $\mathrm{X} \succeq \mathrm{Y}$ if $\mathrm{X} - \mathrm{Y}$ is positive semi-definite (PSD). Let $\mathrm{x} \in \mathbb{R}^{d_x}$ and $\mathrm{y} \in \mathbb{R}^{d_y}$ denote two sets of centered random variables jointly distributed as $\mathscr{D}$ with corresponding auto-covariance matrices $\mathrm{C}_x = \mathbb{E}_\mathrm{x}[\mathrm{xx}^\top]$, $\mathrm{C}_y = \mathbb{E}_\mathrm{y}[\mathrm{yy}^\top]$, and cross-covariance matrix $\mathrm{C}_{xy} = \mathbb{E}_{(\mathrm{x},\mathrm{y})}[\mathrm{xy}^\top]$, and define $d := \max\{d_x, d_y\}$. Finally, $\mathrm{X} \in \mathbb{R}^{d_x \times n}$ and $\mathrm{Y} \in \mathbb{R}^{d_y \times n}$ denote data matrices with $n$ corresponding samples from view 1 and view 2, respectively.

## 1.2 Problem Formulation

Given paired samples $(\mathrm{x}_1, \mathrm{y}_1), \ldots, (\mathrm{x}_T, \mathrm{y}_T)$, drawn i.i.d. from $\mathscr{D}$, the goal is to find a maximally correlated subspace of $\mathscr{D}$, i.e. in terms of the population objective. A simple change of variables in Problem (1), with $\mathrm{U} = \mathrm{C}_x^{1/2}\tilde{\mathrm{U}}$ and $\mathrm{V} = \mathrm{C}_y^{1/2}\tilde{\mathrm{V}}$, yields the following equivalent problem:

$$\text{maximize Tr}\left(\mathrm{U}^\top \mathrm{C}_x^{-\frac{1}{2}} \mathrm{C}_{xy} \mathrm{C}_y^{-\frac{1}{2}} \mathrm{V}\right) \text{ s.t. } \mathrm{U}^\top \mathrm{U} = \mathrm{I}, \ \mathrm{V}^\top \mathrm{V} = \mathrm{I}. \qquad (2)$$

To ensure that Problem 2 is well-posed, we assume that $r := \min\{r_x, r_y\} > 0$, where $r_x = \lambda_{\min}(\mathrm{C}_x)$ and $r_y = \lambda_{\min}(\mathrm{C}_y)$ are smallest eigenvalues of the population auto-covariance matrices. Furthermore, we assume that with probability one, for $(\mathrm{x}, \mathrm{y}) \sim \mathscr{D}$, we have that $\max\{\|\mathrm{x}\|^2, \|\mathrm{y}\|^2\} \leq B$. Let $\Phi \in \mathbb{R}^{d_x \times k}$ and $\Psi \in \mathbb{R}^{d_y \times k}$ denote the top-$k$ left and right singular vectors, respectively, of the population cross-covariance matrix of the whitened views $\mathrm{T} := \mathrm{C}_x^{-1/2}\mathrm{C}_{xy}\mathrm{C}_y^{-1/2}$. It is easy to check that the optimum of Problem (1) is achieved at $\mathrm{U}_* = \mathrm{C}_x^{-1/2}\Phi$, $\mathrm{V}_* = \mathrm{C}_y^{-1/2}\Psi$. Therefore, a natural

approach, given a training dataset, is to estimate empirical auto-covariance and cross-covariance matrices to compute $\widehat{T}$, an empirical estimate of $T$; matrices $U_*$ and $V_*$ can then be estimated using the top-$k$ left and right singular vectors of $\widehat{T}$. This approach is referred to as sample average approximation (SAA) or empirical risk minimization (ERM).

In this paper, we consider the following equivalent re-parameterization of Problem (2) given by the variable substitution $M = UV^\top$, also referred to as lifting. Find $M \in \mathbb{R}^{d_x \times d_y}$ that

$$\text{maximize } \langle M, C_x^{-\frac{1}{2}} C_{xy} C_y^{-\frac{1}{2}} \rangle \text{ s.t. } \sigma_i(M) \in \{0,1\}, i = 1, \ldots, \min\{d_x, d_y\}, \text{ rank}(M) \leq k. \quad (3)$$

We are interested in designing SA algorithms that, for any bounded distribution $\mathscr{D}$ with minimum eigenvalue of the auto-covariance matrices bounded below by $r$, are guaranteed to find an $\epsilon$-suboptimal solution on the population objective (3), from which, we can extract a good solution for Problem (1).

## 1.3 Related Work

There has been a flurry of recent work on scalable approaches to the empirical CCA problem, i.e. methods for numerical optimization of the empirical CCA objective on a fixed data set [1, 8, 14, 15, 24]. These are typically batch approaches which use the entire data set at each iteration, either for performing a power iteration [1, 8] or for optimizing the alternative empirical objective [14, 15, 24]:

$$\text{minimize } \frac{1}{2n} \|\tilde{U}^\top X - \tilde{V}^\top Y\|_F^2 + \lambda_x \|\tilde{U}\|_F^2 + \lambda_y \|\tilde{V}\|_F^2 \text{ s.t. } \tilde{U}^\top C_{x,n} \tilde{U} = I, \ \tilde{V}^\top C_{y,n} \tilde{V} = I, \quad (4)$$

where $C_{x,n}$ and $C_{y,n}$ are the empirical estimates of covariance matrices for the $n$ samples stacked in the matrices $X \in \mathbb{R}^{d_x \times n}$ and $Y \in \mathbb{R}^{d_y \times n}$, using alternating least squares [14], projected gradient descent (AppaGrad, [15]) or alternating SVRG combined with shift-and-invert pre-conditioning [24].

However, all the works above focus only on the empirical problem, and can all be seen as instances of SAA (ERM) approach to the stochastic optimization (learning) problem (1). In particular, the analyses in these works bounds suboptimality on the training objective, not the population objective (1).

The only relevant work we are aware of that studies algorithms for CCA as a population problem is a parallel work by [7]. However, there are several key differences. First, the objective considered in [7] is different from ours. The focus in [7] is on finding a solution $U, V$ that is very similar (has high alignment with) the optimal population solution $U_*, V_*$. In order for this to be possible, [7] must rely on an "eigengap" between the singular values of the cross-correlation matrix $C_{xy}$. In contrast, since we are only concerned with finding a solution that is good in terms of the population objective (2), we need not, and do not, depend on such an eigengap. If there is no eigengap in the cross-correlation matrix, the population optimal solution is not well-defined, but that is fine for us – we are happy to return any optimal (or nearly optimal) solution.

Furthermore, given such an eigengap, the emphasis in [7] is on the guaranteed overall runtime of their method. Their core algorithm is very efficient in terms of runtime, but is not a streaming algorithm and cannot be viewed as an SA algorithm. They do also provide a streaming version, which is runtime and memory efficient, but is still not a "natural" SA algorithm, in that it does not work by making a small update to the solution at each iteration. In contrast, here we present a more "natural" SA algorithm and put more emphasis on its iteration complexity, i.e. the number of samples processed. We do provide polynomial runtime guarantees, but rely on a heuristic capping in order to achieve good runtime performance in practice.

Finally, [7] only consider obtaining the top correlated direction ($k = 1$) and it is not clear how to extend their approach to Problem (1) of finding the top $k \geq 1$ correlated directions. Our methods handle the general problem, with $k \geq 1$, naturally and all our guarantees are valid for any number of desired directions $k$.

## 1.4 Contributions

The goal in this paper is to directly optimize the CCA "population objective" based on i.i.d. draws from the population rather than capturing the *sample*, i.e. the training objective. This view justifies and favors stochastic approximation approaches that are far from optimal on the sample but are essentially as good as the sample average approximation approach on the population. Such a view

has been advocated in supervised machine learning [6, 18]; here, we carry over the same view to the rich world of unsupervised learning. The main contributions of the paper are as follows.

- We give a convex relaxation of the CCA optimization problem. We present two stochastic approximation algorithms for solving the resulting problem. These algorithms work in a streaming setting, i.e. they process one sample at a time, requiring only a single pass through the data, and can easily scale to large datasets.

- The proposed algorithms are instances of inexact stochastic mirror descent with the choice of potential function being Frobenius norm and von Neumann entropy, respectively. Prior work on inexact proximal gradient descent suggests a lower bound on the size of the noise required to guarantee convergence for inexact updates [16]. While that condition is violated here for the CCA problem, we give a tighter analysis of our algorithms with noisy gradients establishing sub-linear convergence rates.

- We give precise iteration complexity bounds for our algorithms, i.e. we give upper bounds on iterations needed to guarantee a user-specified $\epsilon$-suboptimality (w.r.t. population) for CCA. These bounds do not depend on the eigengap in the cross-correlation matrix. To the best of our knowledge this is a first such characterization of CCA in terms of generalization.

- We show empirically that the proposed algorithms outperform existing state-of-the-art methods for CCA on a real dataset. We make our implementation of the proposed algorithms and existing competing techniques available online[1].

## 2 Matrix Stochastic Gradient for CCA (MSG-CCA)

Problem (3) is a non-convex optimization problem, however, it admits a simple convex relaxation. Taking the convex hull of the constraint set in Problem 3 gives the following convex relaxation:

$$\text{maximize } \langle \mathrm{M}, \mathrm{C}_x^{-\frac{1}{2}} \mathrm{C}_{xy} \mathrm{C}_y^{-\frac{1}{2}} \rangle \text{ s.t. } \|\mathrm{M}\|_2 \leq 1, \ \|\mathrm{M}\|_* \leq k. \tag{5}$$

While our updates are designed for Problem (5), our algorithm returns a rank-$k$ solution, through a simple rounding procedure ([27, Algorithm 4]; see more details below), which has the same objective in expectation. This allows us to guarantee $\epsilon$-suboptimality of the output of the algorithm on the original non-convex Problem (3), and equivalently Problem (2).

Similar relaxations have been considered previously to design stochastic approximation (SA) algorithms for principal component analysis (PCA) [2] and partial least squares (PLS) [4]. These SA algorithms are instances of stochastic gradient descent – a popular choice for convex learning problems. However, designing similar updates for the CCA problem is challenging since the gradient of the CCA objective (see Problem (5)) w.r.t. M is $\mathrm{g} := \mathrm{C}_x^{-1/2} \mathrm{C}_{xy} \mathrm{C}_y^{-1/2}$, and it is not at all clear how one can design an unbiased estimator, $\mathrm{g}_t$, of the gradient $\mathrm{g}$ unless one knows the marginal distributions of x and y. Therefore, we consider an instance of *inexact* proximal gradient method [16] which requires access to a first-order oracle with noisy estimates, $\partial_t$, of $\mathrm{g}_t$. We show that an oracle with bound on $\mathbb{E}[\sum_{t=1}^{T} \|\mathrm{g}_t - \partial_t\|]$ of $O(\sqrt{T})$ ensures convergence of the proximal gradient method. Furthermore, we propose a first order oracle with the above property which instantiates the inexact gradient as

$$\partial_t := \mathrm{W}_{x,t} \mathrm{x}_t \mathrm{y}_t^\top \mathrm{W}_{y,t} \approx \mathrm{g}_t, \tag{6}$$

where $\mathrm{W}_{x,t}, \mathrm{W}_{y,t}$ are empirical estimates of whitening transformation based on training data seen until time $t$. This leads to the following stochastic inexact gradient update:

$$\mathrm{M}_{t+1} = \mathscr{P}_F(\mathrm{M}_t + \eta_t \partial_t), \tag{7}$$

where $\mathscr{P}_F$ is the projection operator onto the constraint set of Problem (5).

Algorithm 1 provides the pseudocode for the proposed method which we term inexact matrix stochastic gradient method for CCA (MSG-CCA). At each iteration, we receive a new sample $(\mathrm{x}_t, \mathrm{y}_t)$, update the empirical estimates of the whitening transformations which define the inexact gradient $\partial_t$. This is followed by a gradient update with step-size $\eta$, and projection onto the set of constraints of Problem (5) with respect to the Frobenius norm through the operator $\mathscr{P}_F(\cdot)$ [2]. After $T$ iterations, the algorithm returns a rank-$k$ matrix after a simple rounding procedure [27].

**Algorithm 1** Matrix Stochastic Gradient for CCA (MSG-CCA)
---
**Input:** Training data $\{(\mathbf{x}_t, \mathbf{y}_t)\}_{t=1}^T$, step size $\eta$, auxiliary training data $\{(\mathbf{x}_i', \mathbf{y}_i')\}_{i=1}^\tau$

**Output:** $\tilde{M}$

1: Initialize: $M_1 \leftarrow 0$, $C_{x,0} \leftarrow \frac{1}{\tau}\sum_{i=1}^\tau \mathbf{x}_i'\mathbf{x}_i'^\top$, $C_{y,0} \leftarrow \frac{1}{\tau}\sum_{i=1}^\tau \mathbf{y}_i'\mathbf{y}_i'^\top$
2: **for** $t = 1, \cdots, T$ **do**
3: $\quad$ $C_{x,t} \leftarrow \frac{t+\tau-1}{t+\tau}C_{x,t-1} + \frac{1}{t+\tau}\mathbf{x}_t\mathbf{x}_t^\top$, $W_{x,t} \leftarrow C_{x,t}^{-\frac{1}{2}}$
4: $\quad$ $C_{y,t} \leftarrow \frac{t+\tau-1}{t+\tau}C_{y,t-1} + \frac{1}{t+\tau}\mathbf{y}_t\mathbf{y}_t^\top$, $W_{y,t} \leftarrow C_{y,t}^{-\frac{1}{2}}$
5: $\quad$ $\partial_t \leftarrow W_{x,t}\mathbf{x}_t\mathbf{y}_t^\top W_{y,t}$
6: $\quad$ $M_{t+1} \leftarrow \mathscr{P}_F(M_t + \eta\partial_t)$ $\qquad$ % Projection given in [2]
7: **end for**
8: $\bar{M} = \frac{1}{T}\sum_{t=1}^T M_t$
9: $\tilde{M} = \texttt{rounding}(\bar{M})$ $\qquad$ % Algorithm 2 in [27]

---

We denote the empirical estimates of auto-covariance matrices based on the first $t$ samples by $C_{x,t}$ and $C_{y,t}$. Our analysis of MSG-CCA follows a two-step procedure. First, we show that the empirical estimates of the whitening transform matrices, i.e. $W_{x,t} := C_{x,t}^{-1/2}$, $W_{y,t} := C_{y,t}^{-1/2}$, guarantee that the expected error in the "inexact" estimate, $\partial_t$, converges to zero as $O(1/\sqrt{t})$. Next, we show that the resulting noisy stochastic gradient method converges to the optimum as $O(1/\sqrt{T})$. In what follows, we will denote the true whitening transforms by $W_x := C_x^{-1/2}$ and $W_y := C_y^{-1/2}$.

Since Algorithm 1 requires inverting empirical auto-covariance matrices, we need to ensure that the smallest eigenvalues of $C_{x,t}$ and $C_{y,t}$ are bounded away from zero. Our first technical result shows that in this happens with high probability for all iterates.

**Lemma 2.1.** *With probability* $1-\delta$ *with respect to training data drawn i.i.d. from $\mathscr{D}$, it holds uniformly for all $t$ that* $\lambda_{\min}(C_{x,t}) \geq \frac{r_x}{2}$ *and* $\lambda_{\min}(C_{y,t}) \geq \frac{r_y}{2}$ *whenever:*

$$\tau \geq \max\{\frac{1}{c_x}\log\left(\frac{2d_x}{\log\left(\frac{1}{1-\delta}\right)}\right) - 1, \frac{1}{c_x}\log\left(2d_x\right), \frac{1}{c_y}\log\left(\frac{2d_y}{\log\left(\frac{1}{1-\delta}\right)}\right) - 1, \frac{1}{c_y}\log\left(2d_y\right)\}.$$

*Here* $c_x = \frac{3r_x^2}{6B^2+Br_x}, c_y = \frac{3r_y^2}{6B^2+Br_y}$.

We denote by $\mathscr{A}_t$ the event that for all $j = 1,..,t-1$ the empirical cross-covariance matrices $C_{x,j}$ and $C_{y,j}$ have their smallest eigenvalues bounded from below by $r_x$ and $r_y$, respectively. Lemma 2.1 above, guarantees that this event occurs with probability at least $1-\delta$, as long as there are $\tau = \Omega\left(\frac{B^2}{r^2}\log\left(\frac{2d}{\log\left(\frac{1}{1-\delta}\right)}\right)\right)$ samples in the auxiliary dataset.

**Lemma 2.2.** *Assume that the event $\mathscr{A}_t$ occurs, and that with probability one, for $(\mathbf{x}, \mathbf{y}) \sim \mathscr{D}$, we have* $\max\{\|\mathbf{x}\|^2, \|\mathbf{y}\|^2\} \leq B$. *Then, for* $\kappa := \frac{8B^2\sqrt{2\log(d)}}{r^2}$, *the following holds for all $t$:*

$$\mathbb{E}_{\mathscr{D}}\left[\|\mathbf{g}_t - \partial_t\|_2 \mid \mathscr{A}_t\right] \leq \frac{\kappa}{\sqrt{t}}.$$

The result above bounds the size of the expected noise in the estimate of the inexact gradient. Not surprisingly, the error decays as our estimates of the whitening transformation improve with more data. Moreover, the rate at which the error decreases is sufficient to bound the suboptimality of the MSG-CCA algorithm even with noisy biased stochastic gradients.

**Theorem 2.3.** *After $T$ iterations of MSG-CCA (Algorithm 1) with step size* $\eta = \frac{2\sqrt{k}}{G\sqrt{T}}$, *auxiliary sample of size* $\tau = \Omega(\frac{B^2}{r^2}\log(\frac{2d}{\log(\frac{\sqrt{T}}{\sqrt{T}-1})}))$, *and initializing $M_1 = 0$, the following holds:*

$$\langle M_*, C_x^{-\frac{1}{2}}C_{xy}C_y^{-\frac{1}{2}}\rangle - \mathbb{E}[\langle\tilde{M}, C_x^{-\frac{1}{2}}C_{xy}C_y^{-\frac{1}{2}}\rangle] \leq \frac{2\sqrt{k}G + 2k\kappa + kB/r}{\sqrt{T}}, \qquad (8)$$

*where the expectation is with respect to the i.i.d. samples and rounding, $\kappa$ is as defined in Lemma 2.2,* $\mathrm{M}_*$ *is the optimum of (3),* $\tilde{\mathrm{M}}$ *is the rank-$k$ output of MSG-CCA, and $G = \frac{2B}{\sqrt{r_x r_y}}$.*

While Theorem 2.3 gives a bound on the objective of Problem (3), it implies a bound on the original CCA objective of Problem (1). In particular, given a rank-$k$ factorization of $\tilde{\mathrm{M}} := \mathrm{U}\mathrm{V}^\top$, such that $\mathrm{U}^\top \mathrm{U} = \mathrm{I}_k$ and $\mathrm{V}^\top \mathrm{V} = \mathrm{I}_k$, we construct

$$\tilde{\mathrm{U}} = \mathrm{C}_{x,T}^{-\frac{1}{2}}\mathrm{U}, \quad \tilde{\mathrm{V}} := \mathrm{C}_{y,T}^{-\frac{1}{2}}\mathrm{V}. \tag{9}$$

We then have the following generalization bound.

**Theorem 2.4.** *After $T$ iterations of MSG-CCA (Algorithm 1) with step size $\eta = \frac{2\sqrt{k}}{G\sqrt{T}}$, auxiliary sample of size $\tau = \Omega(\frac{B^2}{r^2}\log(\frac{2d}{\log(\frac{T}{T-1})}))$, and initializing $\mathrm{M}_1 = 0$, the following holds*

$$\mathrm{Tr}(\mathrm{U}_*^\top \mathrm{C}_{xy}\mathrm{V}_*) - \mathbb{E}[\mathrm{Tr}(\tilde{\mathrm{U}}^\top \mathrm{C}_{xy}\tilde{\mathrm{V}})] \leq \frac{2\sqrt{k}G + 2k\kappa}{\sqrt{T}} + \frac{kB}{rT} + \frac{2kB}{r^2}\left(\sqrt{\frac{2B^2}{T}\log(d)} + \frac{2B}{3T}\log(d)\right),$$

$$\mathbb{E}[\|\tilde{\mathrm{U}}^\top \mathrm{C}_x \tilde{\mathrm{U}} - \mathrm{I}\|_2] \leq \frac{B}{r_x^2}\left(\sqrt{\frac{2B^2}{T}\log(d_x)} + \frac{2B}{3T}\log(d_x)\right) + \frac{B+1}{T},$$

$$\mathbb{E}[\|\tilde{\mathrm{V}}^\top \mathrm{C}_y \tilde{\mathrm{V}} - \mathrm{I}\|_2] \leq \frac{B}{r_y^2}\left(\sqrt{\frac{2B^2}{T}\log(d_y)} + \frac{2B}{3T}\log(d_y)\right) + \frac{B+1}{T},$$

*where the expectation is with respect to the i.i.d. samples and rounding, the pair $(\mathrm{U}_*, \mathrm{V}_*)$ is the optimum of (1), $(\tilde{U}, \tilde{V})$ are the factors (defined in (9)) of the rank-$k$ output of MSG-CCA, $r := \min\{r_x, r_y\}$, $d := \max\{d_x, d_y\}$, $\kappa$ is as given in Lemma 2.2, and $G = \frac{2B}{\sqrt{r_x r_y}}$.*

All proofs are deferred to the Appendix in the supplementary material. Few remarks are in order.

**Convexity:** In our design and analysis of MSG-CCA, we have leveraged the following observations: (i) since the objective is linear, an optimum of (5) is always attained at an extreme point, corresponding to an optimum of (3); (ii) the exact convex relaxation (5) is tractable (this is not often the case for non-convex problems); and (iii) although (5) might also have optima not on extreme points, we have an efficient randomized method, called `rounding`, to extract from any feasible point of (5) a solution of (3) that has the same value in expectation [27].

**Eigengap free bound:** Theorem 2.3 and 2.4 do not require an eigengap in the cross-correlation matrix $\mathrm{C}_{xy}$, and in particular the error bound, and thus the implied iteration complexity to achieve a desired suboptimality does not depend on an eigengap.

**Comparison with [7]:** It is not straightforward to compare with the results of [7]. As discussed in Section 1.3, authors in [7] consider only the case $k = 1$ and their objective is different than ours. They seek $(\mathrm{u}, \mathrm{v})$ that have high alignment with the optimal $(\mathrm{u}_*, \mathrm{v}_*)$ as measured through the alignment $\Delta(\bar{\mathrm{u}}, \bar{\mathrm{v}}) := \frac{1}{2}(\bar{\mathrm{u}}^\top \mathrm{C}_x \mathrm{u}_* + \bar{\mathrm{v}}^\top \mathrm{C}_y \mathrm{v}_*)$. Furthermore, the analysis in [7] is dependent on the eigengap $\gamma = \sigma_1 - \sigma_2$ between the top two singular values $\sigma_1, \sigma_2$ of the population cross-correlation matrix $\mathrm{T}$. Nevertheless, one can relate their objective $\Delta(\mathrm{u}, \mathrm{v})$ to ours and ask what their guarantees ensure in terms of our objective, namely achieving $\epsilon$-suboptimality for Problem (3). For the case $k = 1$, and in the presence of an eigengap $\gamma$, the method of [7] can be used to find an $\epsilon$-suboptimal solution to Problem (3) with $O(\frac{\log^2(d)}{\epsilon\gamma^2})$ samples.

**Capped MSG-CCA:** Although MSG-CCA comes with good theoretical guarantees, the computational cost per iteration can be $O(d^3)$. Therefore, we consider a practical variant of MSG-CCA that explicitly controls the rank of the iterates. To ensure computational efficiency, we recommend imposing a hard constraint on the rank of the iterates of MSG-CCA, following an approach similar to previous works on PCA [2] and PLS [4]:

$$\text{maximize } \langle \mathrm{M}, \mathrm{C}_x^{-\frac{1}{2}}\mathrm{C}_{xy}\mathrm{C}_y^{-\frac{1}{2}}\rangle \text{ s.t. } \|\mathrm{M}\|_2 \leq 1, \|\mathrm{M}\|_* \leq k, \mathrm{rank}(\mathrm{M}) \leq K. \tag{10}$$

For estimates of the whitening transformations, at each iteration, we set the smallest $d - K$ eigenvalues of the covariance matrices to a constant (of the order of the estimated smallest eigenvalue of the

covariance matrix). This allows us to efficiently compute the whitening transformations since the covariance matrices decompose into a sum of a low-rank matrix and a scaled identity matrix, bringing down the computational cost per iteration to $O(dK^2)$. We observe empirically on a real dataset (see Section 4) that this procedure along with capping the rank of MSG iterates does not hurt the convergence of MSG-CCA.

# 3 Matrix Exponentiated Gradient for CCA (MEG-CCA)

In this section, we consider matrix multiplicative weight updates for CCA. Multiplicative weights method is a generic algorithmic technique in which one updates a distribution over a set of interest by iteratively multiplying probability mass of elements [12]. In our setting, the set is that of $d$ $k$-dimensional (paired) subspaces and the multiplicative algorithm is an instance of matrix exponentiated gradient (MEG) update. A motivation for considering MEG is the fact that for related problems, including principal component analysis (PCA) and partial least squares (PLS), MEG has been shown to yield fast optimistic rates [4, 22, 26]. Unfortunately we are not able to recover such optimistic rates for CCA as the error in the inexact gradient decreases too slowly.

Our development of MEG requires the symmetrization of Problem (3). Recall that $\mathrm{g} := \mathrm{C}_x^{-1/2}\mathrm{C}_{xy}\mathrm{C}_y^{-1/2}$. Consider the following symmetric matrix $\mathrm{C} := \begin{bmatrix} 0 & \mathrm{g} \\ \mathrm{g} & 0 \end{bmatrix}$ of size $d \times d$, where $d = d_x + d_y$. The matrix $\mathrm{C}$ is referred to as the self-adjoint dilation of the matrix $\mathrm{g}$ [20]. Given the SVD of $\mathrm{g} = \mathrm{U\Sigma V}^\top$ with no repeated singular values, the eigen-decomposition of $\mathrm{C}$ is given as

$$\mathrm{C} = \frac{1}{2} \begin{pmatrix} \mathrm{U} & \mathrm{U} \\ \mathrm{V} & -\mathrm{V} \end{pmatrix} \begin{pmatrix} \Sigma & 0 \\ 0 & -\Sigma \end{pmatrix} \begin{pmatrix} \mathrm{U} & \mathrm{U} \\ \mathrm{V} & -\mathrm{V} \end{pmatrix}^\top.$$

In other words, the top-$k$ left and right singular vectors of $\mathrm{C}_x^{-1/2}\mathrm{C}_{xy}\mathrm{C}_y^{-1/2}$, which comprise the CCA solution we seek, are encoded in top and bottom rows, respectively, of the top-$k$ eigenvectors of its dilation. This suggests the following scaled re-parameterization of Problem (3): find $\mathrm{M} \in \mathbb{R}^{d \times d}$ that

$$\text{maximize } \langle \mathrm{M}, \mathrm{C} \rangle \text{ s.t. } \lambda_i(\mathrm{M}) \in \{0,1\}, i = 1, \ldots, d, \text{ rank}(\mathrm{M}) = k. \tag{11}$$

As in Section 2, we take the convex hull of the constraint set to get a convex relaxation to Problem (11).

$$\text{maximize } \langle \mathrm{M}, \mathrm{C} \rangle \text{ s.t. } \mathrm{M} \succeq 0, \|\mathrm{M}\|_2 \leq 1, \text{Tr}(\mathrm{M}) = k. \tag{12}$$

Stochastic mirror descent on Problem (12) with the choice of potential function being the quantum relative entropy gives the following updates [4, 27]:

$$\widehat{\mathrm{M}}_t = \frac{\exp\left(\log(\mathrm{M}_{t-1}) + \eta\mathrm{C}_t\right)}{\text{Tr}\left(\exp\left(\log(\mathrm{M}_{t-1}) + \eta\mathrm{C}_t\right)\right)}, \quad \mathrm{M}_t = \mathscr{P}\left(\widehat{\mathrm{M}}_t\right), \tag{13}$$

where $\mathrm{C}_t$ is the self-adjoint dilation of unbiased instantaneous gradient $\mathrm{g}_t$, and $\mathscr{P}$ denotes the Bregman projection [10] onto the convex set of constraints in Problem (12). As discussed in Section 2 we only need an inexact gradient estimate $\tilde{\mathrm{C}}_t$ of $\mathrm{C}_t$ with a bound on $\mathbb{E}[\sum_{t=1}^{T} \|\mathrm{C}_t - \tilde{\mathrm{C}}_t\| \mathscr{A}_T]$ of $O(\sqrt{T})$. Setting $\tilde{\mathrm{C}}_t$ to be the self-adjoint dilation of $\partial_t$, defined in Section 2, guarantees such a bound.

**Lemma 3.1.** *Assume that the event $\mathscr{A}_t$ occurs, $\mathrm{g}_t - \partial_t$ has no repeated singular values and that with probability one, for $(\mathrm{x}, \mathrm{y}) \sim \mathscr{D}$, we have $\max\{\|\mathrm{x}\|^2, \|\mathrm{y}\|^2\} \leq B$. Then, for $\kappa$ defined in lemma 2.2, we have that, $\mathbb{E}_{\mathrm{x}_t, \mathrm{y}_t}\langle \mathrm{M}_{t-1} - \mathrm{M}_*, \mathrm{C}_t - \tilde{\mathrm{C}}_t | \mathscr{A}_t\rangle \leq \frac{2k\kappa}{\sqrt{t}}$, where $\mathrm{M}_*$ is the optimum of Problem (11).*

Using the bound above, we can bound the suboptimality gap in the population objective between the true rank-$k$ CCA solution and the rank-$k$ solution returned by MEG-CCA.

**Theorem 3.2.** *After $T$ iterations of MEG-CCA (see Algorithm 2 in Appendix) with step size $\eta = \frac{1}{G}\log\left(1 + \sqrt{\frac{\log(d)}{GT}}\right)$, auxiliary sample of size $\tau = \Omega(\frac{B^2}{r^2}\log(\frac{2d}{\log(\frac{\sqrt{T}}{\sqrt{T}-1})}))$ and initializing $\mathrm{M}_0 = \frac{1}{d}\mathrm{I}$, the following holds:*

$$\langle \mathrm{M}_*, \mathrm{C} \rangle - \mathbb{E}[\langle \tilde{\mathrm{M}}, \mathrm{C} \rangle] \leq 2k\sqrt{\frac{G^2\log(d)}{T}} + 2\frac{k\kappa}{\sqrt{T}},$$

*where the conditional expectation is taken with respect to the distribution and the internal randomization of the algorithm, $M_*$ is the optimum of Problem (11), $\tilde{M}$ is the rank-$k$ output of MEG-CCA after rounding, $G = \frac{2B}{\sqrt{r_x r_y}}$ and $\kappa$ is defined in Lemma 2.2.*

All of our remarks regarding latent convexity of the problem and practical variants from Section 2 apply to MEG-CCA as well. We note, however, that without additional assumptions like eigengap for T we are not able to recover projections to the canonical subspaces as done in Theorem 2.4.

## 4  Experiments

We provide experimental results for our proposed methods, in particular we compare capped-MSG which is the practical variant of Algorithm 1 with capping as defined in equation (10), and MEG (Algorithm 2 in the Appendix), on a real dataset, `Mediamill` [19], consisting of paired observations of videos and corresponding commentary. We compare our algorithms against CCALin of [8], ALS CCA of [24][2], and SAA, which is denoted by "batch" in Figure 1. All of the comparisons are given in terms of the CCA objective as a function of either CPU runtime or number of iterations. The target dimensionality in our experiments is $k \in \{1, 2, 4\}$. The choice of $k$ is dictated largely by the fact that the spectrum of the `Mediamill` dataset decays exponentially. To ensure that the problem is well-conditioned, we add $\lambda I$ for $\lambda = 0.1$ to the empirical estimates of the covariance matrices on `Mediamill` dataset. For both MSG and MEG we set the step size at iteration $t$ to be $\eta_t = \frac{0.1}{\sqrt{t}}$.

`Mediamill` is a multiview dataset consisting of $n = 10,000$ corresponding videos and text annotations with labels representing semantic concepts [19]. The image view consists of 120-dimensional visual features extracted from representative frames selected from videos, and the textual features are 100-dimensional. We give the competing algorithms, both CCALin and ALS CCA, the advantage of the knowledge of the eigengap at $k$. In particular, we estimate the spectrum of the matrix $\widehat{T}$ for the `Mediamill` dataset and set the gap-dependent parameters in CCALin and ALS CCA accordingly. We note, however, that estimating the eigengap to set the parameters is impractical in real scenarios. Both CCALin and ALS CCA will therefore require additional tuning compared to MSG and MEG algorithms proposed here. In the experiments, we observe that CCALin and ALS CCA outperform MEG and capped-MSG when recovering the top CCA component, in terms of progress per-iteration. However, capped-MSG is the best in terms of the overall runtime. The plots are shown in Figure 1.

## 5  Discussion

We study CCA as a stochastic optimization problem and show that it is efficiently learnable by providing analysis for two stochastic approximation algorithms. In particular, the proposed algorithms achieve $\epsilon$-suboptimality in population objective in iterations $O(\frac{1}{\epsilon^2})$.

Note that both of our Algorithms, MSG-CCA in Algorithm 1 and MEG-CCA in Algorithm 2 in Appendix B are instances of inexact proximal-gradient method which was studied in [16]. In particular, both algorithms receive a noisy gradient $\partial_t = g_t + E_t$ at iteration $t$ and perform exact proximal steps (Bregman projections in equations (7) and (13)). The main result in [16] provides an $O(E^2/T)$ convergence rate, where $E = \sum_{t=1}^{T} \|E_t\|$ is the partial sum of the errors in the gradients. It is shown that $E = o(\sqrt{T})$ is a necessary condition to obtain convergence. However, for the CCA problem that we are considering in this paper, our lemma A.6 shows that $E = O(\sqrt{T})$. In fact, it is easy to see that $E = \Theta(\sqrt{T})$. Our analysis yields $O(\frac{1}{\sqrt{T}})$ convergence rates for both Algorithms 1 and 2. This perhaps warrants further investigation into the more general problem of inexact proximal gradient method.

In empirical comparisons, we found the capped version of the proposed MSG algorithm to outperform other methods including MEG in terms of overall runtime needed to reach an $\epsilon$-suboptimal solution. Future work will focus on gaining a better theoretical understanding of capped MSG.

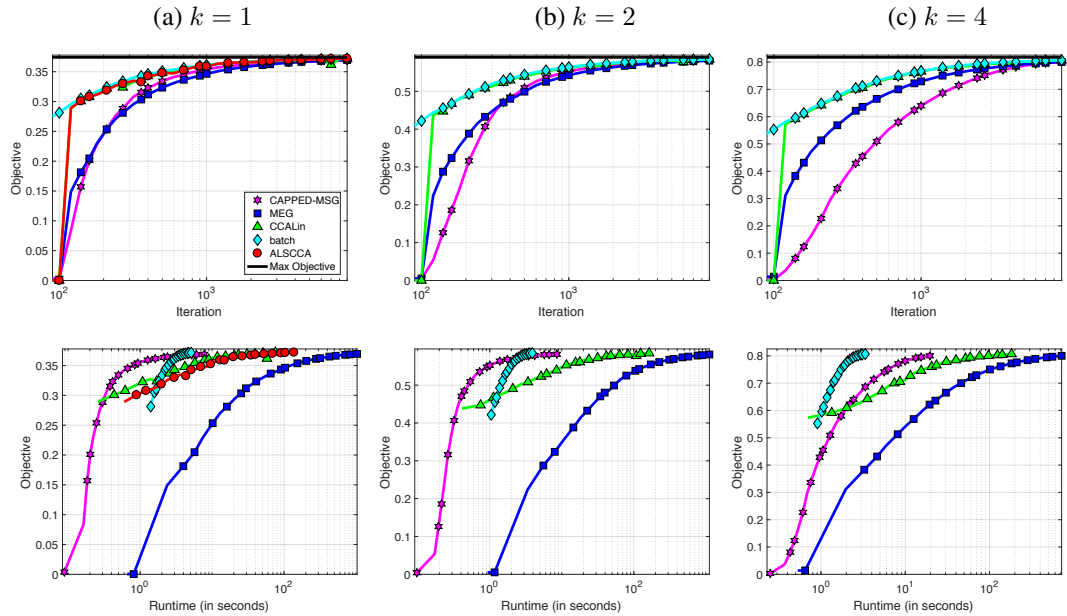

Figure 1: Comparisons of CCA-Lin, CCA-ALS, MSG, and MEG for CCA optimization on the MediaMill dataset, in terms of the objective value as a function of iteration (top) and as a function of CPU runtime (bottom).

## Acknowledgements

This research was supported in part by NSF BIGDATA grant IIS-1546482.

## Footnotes

[1]https://www.dropbox.com/sh/dkz4zgkevfyzif3/AABK9JlUvIUYtHvLPCBXLlpha?dl=0

[2]We run ALS only for $k = 1$ as the algorithm and the current implementation from the authors does not handle $k \geq 1$.

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
