[Supplementary Material · supp-nips2017-2494.pdf]



## A   Matrix Stochastic Gradient for CCA

Throughout this section, we denote the error in the gradient at time $t$ by $E_t = g_t - \partial_t$. First, we introduce the following structural results, which give a lower bound on the smallest eigenvalue of the empirical auto-covariance matrices, which holds with high probability for any iterate (A.2), and uniformly over all iterates (2.1). We will use Matrix Bernstein [20] inequality in proof of Lemma A.2.

**Theorem A.1** (Matrix Bernstein [20]). *consider a finite sequence $\{X_k\}$ of independent, random, self-adjoint matrices with dimension $d$. Assume that each random matrix satisfies $\mathbb{E}[X_k] = 0$ and $\lambda_{\max}(X_k) \leq R$ almost surely. Then, for all $\epsilon \geq 0$,*

$$\mathbb{P}\left\{\lambda_{\max}\left(\sum_k X_k\right) \geq \epsilon\right\} \leq d \exp\left(\frac{-\epsilon^2/2}{\sigma^2 + R\epsilon/3}\right)$$

*where $\sigma^2 := \|\sum_k \mathbb{E}[X_k^2]\|$.*

**Lemma A.2.** *With probability at least $1 - \delta'$ with respect to training data drawn i.i.d. from $\mathscr{D}$, it holds that $\lambda_{\min}(C_{x,\tau}) \geq \frac{r_x}{2}$ and $\lambda_{\min}(C_{y,\tau}) \geq \frac{r_y}{2}$, whenever*

$$\tau \geq \max\left\{\frac{2\log\left(\frac{d_x}{\delta'}\right)B^2}{r_x^2} + \frac{\log\left(\frac{d_x}{\delta'}\right)B}{3r_x}, \frac{2\log\left(\frac{d_y}{\delta'}\right)B^2}{r_y^2} + \frac{\log\left(\frac{d_y}{\delta'}\right)B}{3r_y}\right\}.$$

*Proof of Lemma A.2.* Set $X_k = \frac{1}{t}\left(x_k x_k^\top - C_x\right)$, so that $\mathbb{E}\left[\left\|\sum_{k=1}^t X_k\right\|_2\right] = \mathbb{E}\left[\|C_{x,t} - C_x\|_2\right]$. Define

$$\sigma^2 = \sum_{k=1}^\tau \mathbb{E}\left[X_k^2\right] \preceq \frac{1}{\tau^2}\sum_k \left\{\mathbb{E}\left[Bx_k x_k^\top\right] - C_x^2\right\} \preceq \frac{B}{\tau^2}\mathbb{E}\left[\sum_{k=1}^\tau x_k x_k^\top\right] \leq \frac{B^2}{\tau}.$$

Using Theorem A.1 with $\epsilon = \frac{r_x}{2}$ we get that with probability at least $1 - \delta'$, it holds that $\|C_x - C_{x,t}\| \leq \frac{r_x}{2}$. By Weyl's inequality, we have that

$$|\lambda_{\min}(C_{x,\tau}) - r_x| = |\lambda_{\min}(C_{x,\tau}) - \lambda_{\min}(C_x)| \leq \|C_x - C_{x,\tau}\| \leq \frac{r_x}{2}.$$

A similar derivation for $\lambda_{\min}(C_{y,\tau})$ completes the proof. $\qquad\square$

*Proof of Lemma 2.1.* We show the result for $C_{x,t}$ with $c = c_x = \frac{3r_x^2}{6B^2 + Br_x}$. Proof for $C_{y,t}$ is symmetric. By Lemma A.2, we have that for every $t$:

$$\mathbb{P}\{\|C_x - C_{x,t}\| \leq \frac{r_x}{2}\} \geq 1 - d_x e^{-ct}.$$

Probability that $\lambda_{\min}(C_{x,t}) \geq \frac{r_x}{2}$ uniformly for all $t \geq \tau + 1$ is $\prod_{t=\tau+1}^{T+\tau}\left(1 - d_x e^{-ct}\right)$. Taking the logarithm, we have

$$\log\left(\prod_{t=\tau+1}^{T+\tau}\left(1 - d_x e^{-ct}\right)\right) = \sum_{t=\tau+1}^{T+\tau}\log\left(1 - d_x e^{-ct}\right)$$

$$\geq \sum_{t=\tau+1}^{T+\tau} -2d_x e^{-ct} \qquad (\log\left(1 - z\right) \geq -2z \text{ for } z \in (0, 0.5))$$

$$= -2d_x \frac{(e^{-c})^{\tau+1} - (e^{-c})^{T+\tau+1}}{1 - e^{-c}}$$

$$\geq -2d_x \frac{e^{-c(\tau+1)}}{1 - e^{-c}}$$

where we require $d_x e^{-ct} \leq \frac{1}{2}$, which holds for $\tau \geq \frac{1}{c} \log (2d_x)$. We want $\exp\left(-2d_x \frac{e^{-c(\tau+1)}}{1-e^{-c}}\right) \geq 1 - \delta$, which gives the following

$$\exp\left(-2d_x \frac{e^{-c(\tau+1)}}{1-e^{-c}}\right) \geq 1 - \delta$$

$$\Longleftrightarrow -e^{-c(\tau+1)} \geq \frac{1-e^{-c}}{2d_x} \log (1-\delta)$$

$$\Longleftrightarrow -c(\tau+1) \leq \log\left(\frac{1-e^{-c}}{2d_x} \log\left(\frac{1}{1-\delta}\right)\right)$$

$$\Longleftrightarrow \tau \geq \frac{1}{c} \log\left(\frac{2d_x}{\log\left(\frac{1}{1-\delta}\right)}\right) - 1$$

so that the algorithm succeeds whenever $\tau \geq \max\{\frac{1}{c} \log\left(\frac{1-e^{-c}}{2d_x} \log\left(\frac{1}{1-\delta}\right)\right) - 1, \frac{1}{c} \log (2d_x)\}$. $\qquad\square$

**Remark A.3.** *Throughout the Appendix, $\delta$ and $\tau$ are as defined in statement of Lemma 2.1.*

Next, we introduce a result on perturbations of matrix square roots which is used in proof of Lemma 2.2.

**Lemma A.4** (Perturbation Bounds for Matrix Square Roots [17]). *Let $A_j \in \mathbb{R}^{n \times n}$ with $A_j \succeq \mu_j^2 I$ in the positive semi-definite order where $j = 1, 2$. Then $A_j$ has a square root satisfying $A_j^{\frac{1}{2}} \succeq \mu_j I$ and $\|A_1^{\frac{1}{2}} - A_2^{\frac{1}{2}}\|_2 \leq \frac{1}{\mu_1 + \mu_2} \|A_1 - A_2\|_2$.*

Next, we present the following bound on convergence of the empirical covariance matrix to the population covariance matrix.

**Lemma A.5.** *Under the same assumptions as Lemma 2.2*

$$\mathbb{E}_{\mathcal{D}}\left[\|C_{x,t} - C_x\|_2\right] \leq \sqrt{\frac{2B^2}{t} \log (d_x)} + \frac{B}{3t} \log (d_x).$$

*Proof.* We bound the quantity by applying the Matrix Bernstein Inequality ([21], Theorem 6.6.1). Set $X_k = \frac{1}{t}\left(x_k x_k^\top - C_x\right)$, so that $\mathbb{E}\left[\left\|\sum_{k=1}^{t} X_k\right\|_2\right] = \mathbb{E}\left[\|C_{x,t} - C_x\|_2\right]$. To apply the inequality we need to verify that $\mathbb{E}[X_k] = 0$ and $\|X_k\|_2 \leq R$ and bound $\sigma^2 := \left\|\sum_k \mathbb{E}[X_k^2]\right\|_2$. It follows from the definition that $\mathbb{E}[X_k] = 0$. To bound $\|X_k\|_2$, note that

$$\|X_k\|_2 = \frac{1}{t}\left\|x_k x_k^\top - C_x\right\|_2 \leq \frac{1}{t}\left(\|x_k x_k^\top\|_2 + \|\mathbb{E}[x_k x_k^\top]\|_2\right) \leq \frac{1}{t}\left(B + \mathbb{E}[\|x_k x_k^\top\|_2]\right) \leq \frac{2B}{t}.$$

Finally, we bound $\sigma^2$ by observing that

$$\sum_{k=1}^{t} \mathbb{E}[X_k^2] \preceq \frac{1}{t^2} \sum_k \left\{\mathbb{E}[B x_k x_k^\top] - C_x^2\right\} \preceq \frac{B}{t^2} \mathbb{E}\left[\sum_{k=1}^{t} x_k x_k^\top\right],$$

which implies $\sigma^2 \leq \frac{B^2}{t}$. By Matrix Bernstein's Inequality we have

$$\mathbb{E}\left[\left\|\sum_{k=1}^{t} X_k\right\|_2\right] = \mathbb{E}\left[\|C_{x,t} - C_x\|_2\right] \leq \sqrt{\frac{2B^2}{t} \log (d_x)} + \frac{B}{3t} \log (d_x)$$

which completes the proof. $\qquad\square$

*Proof of lemma 2.2.* Let $A = W_x x_t, B = W_y y_t, \widehat{A} = W_{x,t} x_t, \widehat{B} = W_{y,t} y_t$. From the lower bound assumption on the spectrum of the population auto-covariance matrices and Lemma 2.1 we have

$\frac{1}{\sqrt{r_x}}\mathrm{I} \succeq \mathrm{W}_x, \sqrt{\frac{2}{r_x}}\mathrm{I} \succeq \mathrm{W}_{x,t}, \frac{1}{\sqrt{r_y}}\mathrm{I} \succeq \mathrm{W}_y, \sqrt{\frac{2}{r_y}}\mathrm{I} \succeq \mathrm{W}_{y,t}$. Therefore,

$$
\begin{aligned}
\mathbb{E}\left[\|\mathrm{E}_t\|_2 \,|\mathscr{A}_t\right] = \mathbb{E}\left[\|\mathrm{g}_t - \partial_t\|_2 \,|\mathscr{A}_t\right] &= \mathbb{E}\left[\left\|\mathrm{W}_x\mathrm{x}_t\mathrm{y}_t^\top \mathrm{W}_y - \mathrm{W}_{x,t}\mathrm{x}_t\mathrm{y}_t^\top \mathrm{W}_{y,t}\right\|_2 \,|\mathscr{A}_t\right] \\
&= \mathbb{E}\left[\left\|\mathrm{A}\mathrm{B}^\top - \widehat{\mathrm{A}}\widehat{\mathrm{B}}^\top\right\|_2 \,|\mathscr{A}_t\right] \\
&= \mathbb{E}\left[\left\|\mathrm{A}\mathrm{B}^\top - \mathrm{A}\widehat{\mathrm{B}}^\top + \mathrm{A}\widehat{\mathrm{B}}^\top - \widehat{\mathrm{A}}\widehat{\mathrm{B}}^\top\right\|_2 \,|\mathscr{A}_t\right] \\
&\leq \mathbb{E}\left[\|\mathrm{A}\|_2\left\|\mathrm{B} - \widehat{\mathrm{B}}\right\|_2 + \left\|\widehat{\mathrm{B}}\right\|_2\left\|\mathrm{A} - \widehat{\mathrm{A}}\right\|_2 \,|\mathscr{A}_t\right].
\end{aligned} \tag{14}
$$

where the inequality is due to the triangle inequality and sub-multiplicativity of the operator norm. We first bound $\|\mathrm{A}\|_2$ and $\left\|\widehat{\mathrm{B}}\right\|_2$

$$
\|\mathrm{A}\|_2 \leq \|\mathrm{W}_x\|_2 \|\mathrm{x}_t\| \leq \sqrt{\frac{B}{r_x}}, \quad \left\|\widehat{\mathrm{B}}\right\|_2 \leq \|\mathrm{W}_{y,t}\|_2 \|\mathrm{y}_t\| \leq \sqrt{\frac{2B}{r_y}}.
$$

This implies that (14) is bounded by

$$
\sqrt{\frac{B}{r_x}}\mathbb{E}\left[\left\|\mathrm{B} - \widehat{\mathrm{B}}\right\|_2 \,|\mathscr{A}_t\right] + \sqrt{\frac{2B}{r_y}}\mathbb{E}\left[\left\|\mathrm{A} - \widehat{\mathrm{A}}\right\|_2 \,|\mathscr{A}_t\right]. \tag{15}
$$

We now bound $\mathbb{E}\left[\left\|\mathrm{A} - \widehat{\mathrm{A}}\right\|_2 \,|\mathscr{A}_t\right]$

$$
\begin{aligned}
\mathbb{E}\left[\left\|\mathrm{A} - \widehat{\mathrm{A}}\right\|_2 \,|\mathscr{A}_t\right] &\leq \mathbb{E}\left[\|\mathrm{W}_x - \mathrm{W}_{x,t}\|_2 \|\mathrm{x}_t\| \,|\mathscr{A}_t\right] \\
&\leq \sqrt{B}\,\mathbb{E}\left[\left\|\mathrm{C}_x^{-\frac{1}{2}} - \mathrm{C}_{x,t}^{-\frac{1}{2}}\right\|_2 \,|\mathscr{A}_t\right] \\
&= \sqrt{B}\,\mathbb{E}\left[\left\|\mathrm{C}_x^{-\frac{1}{2}}\left(\mathrm{C}_{x,t}^{\frac{1}{2}} - \mathrm{C}_x^{\frac{1}{2}}\right)\mathrm{C}_{x,t}^{-\frac{1}{2}}\right\|_2 \,|\mathscr{A}_t\right] \\
&\leq \sqrt{B}\,\mathbb{E}\left[\left\|\mathrm{C}_x^{-\frac{1}{2}}\right\|_2 \left\|\mathrm{C}_{x,t}^{-\frac{1}{2}}\right\|_2 \left\|\mathrm{C}_{x,t}^{\frac{1}{2}} - \mathrm{C}_x^{\frac{1}{2}}\right\|_2 \,|\mathscr{A}_t\right] \\
&\leq \frac{\sqrt{2B}}{r_x}\mathbb{E}\left[\left\|\mathrm{C}_{x,t}^{\frac{1}{2}} - \mathrm{C}_x^{\frac{1}{2}}\right\|_2 \,|\mathscr{A}_t\right] \\
&\leq \frac{\sqrt{2B}}{(1+\sqrt{2}/2)r_x^{3/2}}\mathbb{E}\left[\|\mathrm{C}_{x,t} - \mathrm{C}_x\|_2 \,|\mathscr{A}_t\right] \leq \frac{\sqrt{B}}{r_x^{3/2}}\mathbb{E}\left[\|\mathrm{C}_{x,t} - \mathrm{C}_x\|_2 \,|\mathscr{A}_t\right],
\end{aligned} \tag{16}
$$

where the last inequality follows from Lemma A.4. By Lemma A.5 $\mathbb{E}\left[\|\mathrm{C}_{x,t} - \mathrm{C}_x\|_2\right] \leq \sqrt{\frac{2B^2}{t}\log(d_x)} + \frac{2B}{3t}\log(d_x)$ and thus by equation (16),

$$
\begin{aligned}
\mathbb{E}\left[\left\|\mathrm{A} - \widehat{\mathrm{A}}\right\|_2 \,|\mathscr{A}_t\right] &\leq \frac{\sqrt{B}}{r_x^{3/2}}\left\{\sqrt{\frac{2B^2}{t}\log(d_x)} + \frac{B}{3t}\log(d_x)\right\} \\
\mathbb{E}\left[\left\|\mathrm{B} - \widehat{\mathrm{B}}\right\|_2 \,|\mathscr{A}_t\right] &\leq \frac{\sqrt{B}}{r_y^{3/2}}\left\{\sqrt{\frac{2B^2}{t}\log(d_y)} + \frac{B}{3t}\log(d_y)\right\}
\end{aligned} \tag{17}
$$

Finally (15) together with (17) implies that

$$
\mathbb{E}\left[\|\mathrm{E}_t\|_2 \,|\mathscr{A}_t\right] \leq \frac{2B^2}{\sqrt{r_x r_y}}\left\{\frac{1}{r_y}\left\{\sqrt{\frac{2\log(d_y)}{t}} + \frac{3}{t}\log(d_y)\right\} + \frac{1}{r_x}\left\{\sqrt{\frac{2\log(d_x)}{t}} + \frac{3}{t}\log(d_x)\right\}\right\}.
$$

Let $r := \min\{r_x, r_y\}$ and $d := \max\{d_x, d_y\}$. As long as $t > \frac{9\log(d)}{2}$, we have that $\mathbb{E}\left[\|\mathrm{E}_t\|_2 \,|\mathscr{A}_t\right] \leq \frac{\kappa}{\sqrt{t}}$, where $\kappa := \frac{8B^2\sqrt{2\log(d)}}{r^2}$. $\qquad\square$

**Lemma A.6.** *Assume that the event $\mathscr{A}_T$ occurs. Let $\kappa$ be a constant such that for all iterates $\mathbb{E}_{\mathscr{D}}\left[\|\mathrm{g}_t - \partial_t\|_2 |\mathscr{A}_t\right] \leq \frac{\kappa}{\sqrt{t}}$. Then, we have that $\sum_{t=1}^T \mathbb{E}\left[\|\mathrm{E}_t\|_2 |\mathscr{A}_t\right] \leq 2\kappa\sqrt{T}$.*

*Proof.* We note that $\sum_{t=1}^{T} \frac{1}{\sqrt{t}} \leq \int_{t=1}^{T} \frac{1}{\sqrt{t}}\, dt + 1$. Substituting $z = \sqrt{t}$ and noting $dt = 2z\, dz$ we get

$$\int_{t=1}^{T} \frac{1}{\sqrt{t}}\, dt + 1 = \int_{z=1}^{\sqrt{T}} \frac{1}{z}\, 2z\, dz + 1 = 2\sqrt{T} - 1 \leq 2\sqrt{T}.$$

$\square$

**Lemma A.7.** *With the same assumptions as in Lemma 2.2, we have* $\|\partial_t\|_F \leq \frac{2B}{\sqrt{r_x r_y}}$.

*Proof.* $\|\partial_t\|_F = \left\|\mathrm{W}_{x,t}\mathrm{x}_t\mathrm{y}_t^\top \mathrm{W}_{y,t}^\top\right\|_F = \|\mathrm{W}_{x,t}\mathrm{x}_t\|_2 \|\mathrm{W}_{y,t}\mathrm{y}_t\|_2 \leq B\|\mathrm{W}_{x,t}\|_2\|\mathrm{W}_{y,t}\|_2 \leq \frac{2B}{\sqrt{r_x r_y}}.$

$\square$

*Proof of Theorem 2.3.* Analysis is done by conditioning on the events that $\lambda_{\min}(\mathrm{C}_{x,t}) \geq \frac{r_x}{2}$ and $\lambda_{\min}(\mathrm{C}_{y,t}) \geq \frac{r_y}{2}$. By Lemma 2.1 we know that this event occurs with probability at least $1 - \delta$, where $\tau \geq \max\{\frac{1}{c_x}\log\left(\frac{1-e^{-c}}{2d_x}\log(1-\delta)\right) - 1, \frac{1}{c_x}\log(2d_x), \frac{1}{c_y}\log\left(\frac{1-e^{-c}}{2d_y}\log(1-\delta)\right) - 1, \frac{1}{c_y}\log(2d_y)\}$. The expectations are taken by conditioning on the above events and for ease of notation we set $r_x = \frac{r_x}{2}, r_y = \frac{r_y}{2}$. We start the analysis by measuring the distance between the $t$-th iterate and the optimum, $D_t = \|\mathrm{M}_t - \mathrm{M}_*\|_F$.

$$\begin{aligned}
D_{t+1}^2 = \|\mathrm{M}_{t+1} - \mathrm{M}_*\|_F^2 &= \|\mathscr{P}_F(\mathrm{M}_t + \eta\partial_t) - \mathrm{M}_*\|_F^2 \\
&\leq \|\mathrm{M}_t + \eta\partial_t - \mathrm{M}_*\|_F^2 \\
&= \|\mathrm{M}_t - \mathrm{M}_*\|_F^2 + \eta^2\|\partial_t\|_F^2 + 2\eta\langle \mathrm{M}_t - \mathrm{M}_*, \mathrm{g}_t + \mathrm{E}_t\rangle \\
&\leq D_t^2 + \eta^2 G^2 + 2\eta\langle \mathrm{M}_t - \mathrm{M}_*, \mathrm{g}_t\rangle + 2\eta\langle \mathrm{M}_t - \mathrm{M}_*, \mathrm{E}_t\rangle \\
&\leq D_t^2 + \eta^2 G^2 + 2\eta\langle \mathrm{M}_t - \mathrm{M}_*, \mathrm{g}_t\rangle + 2\eta\|\mathrm{M}_t - \mathrm{M}_*\|_*\|\mathrm{E}_t\|_2 \\
&\leq D_t^2 + \eta^2 G^2 + 2\eta\langle \mathrm{M}_t - \mathrm{M}_*, \mathrm{g}_t\rangle + 4k\eta\|\mathrm{E}_t\|_2,
\end{aligned}$$

where the first inequality follows since projection onto a convex set in a Hilbert space is contractive, the second inequality follows since $G = 2B/\sqrt{r_x r_y}$ is an upper bound on $\|\partial_t\|_F$ as given in Lemma A.7, the third inequality follows using Holder's inequality, and the last inequality follows since $\|\mathrm{M}_t - \mathrm{M}_*\|_* \leq \|\mathrm{M}_t\|_* + \|\mathrm{M}_*\|_* \leq 2k$. Rearranging, dividing both sides by $2\eta$, and taking expectation on both sides, we get

$$\mathbb{E}[\langle \mathrm{M}_* - \mathrm{M}_t, \mathrm{g}_t\rangle | \mathscr{A}_t] \leq \frac{D_t^2 - D_{t+1}^2}{2\eta} + \frac{\eta}{2}G^2 + 2k\mathbb{E}[\|\mathrm{E}_t\|_2 | \mathscr{A}_t]$$

where $\mathrm{M}_t$ and $\mathrm{g}_t$ are conditionally independent given $\mathscr{A}_t$. We average over $T$ iterates, and note that $\sum_{t=1}^{T} D_t^2 - D_{t+1}^2 = D_1^2 - D_{T+1}^2 \leq D_1^2$, where the initial distance is bounded as follows:

$$D_1^2 = \|\mathrm{M}_1 - \mathrm{M}_*\|_F^2 = \|\mathrm{M}_1\|_F^2 + \|\mathrm{M}_*\|_F^2 - 2\langle \mathrm{M}_1, \mathrm{M}_*\rangle \leq k + k + 2\|\mathrm{M}_1\|_*\|\mathrm{M}_*\|_2 \leq 4k.$$

We get:

$$\mathbb{E}[\langle \mathrm{M}_* - \tilde{\mathrm{M}}, \mathrm{C}_x^{-\frac{1}{2}}\mathrm{C}_{xy}\mathrm{C}_y^{-\frac{1}{2}}\rangle | \mathscr{A}_T] \leq \frac{2k}{\eta T} + \frac{\eta G^2}{2} + \frac{2k\kappa\sqrt{T}}{T},$$

where we used Lemma A.6 to bound $\sum_{t=1}^{T} \mathbb{E}\left[\|\mathrm{E}_t\|_2 | \mathscr{A}_t\right] \leq 2\kappa\sqrt{T}$. Finally write

$$\begin{aligned}
\mathbb{E}[\langle \mathrm{M}_* - \tilde{\mathrm{M}}, \mathrm{C}_x^{-\frac{1}{2}}\mathrm{C}_{xy}\mathrm{C}_y^{-\frac{1}{2}}\rangle] &= \mathbb{E}[\langle \mathrm{M}_* - \tilde{\mathrm{M}}, \mathrm{C}_x^{-\frac{1}{2}}\mathrm{C}_{xy}\mathrm{C}_y^{-\frac{1}{2}}\rangle | \mathscr{A}_T](1 - \delta) + \mathbb{E}[\langle \mathrm{M}_* - \tilde{\mathrm{M}}, \mathrm{C}_x^{-\frac{1}{2}}\mathrm{C}_{xy}\mathrm{C}_y^{-\frac{1}{2}}\rangle | \bar{\mathscr{A}}_T]\delta \\
&\leq \mathbb{E}[\langle \mathrm{M}_* - \tilde{\mathrm{M}}, \mathrm{C}_x^{-\frac{1}{2}}\mathrm{C}_{xy}\mathrm{C}_y^{-\frac{1}{2}}\rangle | \mathscr{A}_T] + \delta\mathbb{E}[\langle \mathrm{M}_* - \tilde{\mathrm{M}}, \mathrm{C}_x^{-\frac{1}{2}}\mathrm{C}_{xy}\mathrm{C}_y^{-\frac{1}{2}}\rangle | \bar{\mathscr{A}}_T] \\
&\leq \frac{2k}{\eta T} + \frac{\eta G^2}{2} + \frac{2k\kappa\sqrt{T}}{T} + \delta\frac{Bk}{r},
\end{aligned}$$

where the last inequality holds because $\langle \mathrm{M}_*, \mathrm{C}_x^{-\frac{1}{2}}\mathrm{C}_{xy}\mathrm{C}_y^{-\frac{1}{2}}\rangle < \frac{Bk}{r}$. To finish the proof we can set $\delta \leq \frac{1}{\sqrt{T}}$ and choose optimal learning rate $\eta = \frac{2\sqrt{k}}{G\sqrt{T}}$.

$\square$

While Theorem 2.3 gives a bound on the objective of Problem 2, we can always bound the original CCA objective as given in Problem 1. Note that after rounding, we get a rank-$k$ factorization for $\tilde{M} := UV^\top$, such that $U^\top U = I_k$ and $V^\top V = I_k$. As a result, for $\widehat{U} := C_x^{-\frac{1}{2}} U$ and $\widehat{V} := C_y^{-\frac{1}{2}} V$ it holds that $\widehat{U}^\top C_x \widehat{U} = \widehat{V}^\top C_y \widehat{V} = I_k$. Furthermore, it holds that:

$$\text{Tr}(U_*^\top C_{xy} V_* - \widehat{U}^\top C_{xy} \widehat{V}) \leq \frac{2\sqrt{k}G + 2k\kappa}{\sqrt{T}}$$

Let's denote $\tilde{U} := C_{x,t}^{-\frac{1}{2}} U$ and $\tilde{V} := C_{y,t}^{-\frac{1}{2}} V$. We first give the following structural lemma which is used in Theorem 2.4 for giving generalization error bounds for $\tilde{U}$ and $\tilde{V}$ with respect to the original CCA problem as in 1.

**Lemma A.8.** *Assume the event $\mathscr{A}_T$ occurs, then:*

$$\mathbb{E}[\|\tilde{U} - \widehat{U}\|_2 | \mathscr{A}_T] = \mathbb{E}\left[\left\|C_x^{-\frac{1}{2}} - C_{x,T}^{-\frac{1}{2}}\right\|_2 \Big| \mathscr{A}_T\right] \leq \frac{1}{r_x^{3/2}} \left( \sqrt{\frac{2B^2}{T} \log(d_x)} + \frac{2B}{3T} \log(d_x) \right)$$

$$\mathbb{E}[\|\tilde{V} - \widehat{V}\|_2 | \mathscr{A}_T] = \mathbb{E}\left[\left\|C_y^{-\frac{1}{2}} - C_{y,T}^{-\frac{1}{2}}\right\|_2 \Big| \mathscr{A}_T\right] \leq \frac{1}{r_y^{3/2}} \left( \sqrt{\frac{2B^2}{T} \log(d_y)} + \frac{2B}{3T} \log(d_y) \right)$$

*Proof.* First observe that

$$\mathbb{E}[\|\tilde{U} - \widehat{U}\|_2 | \mathscr{A}_T] = \mathbb{E}[\|C_{x,T}^{-\frac{1}{2}} U - C_x^{-\frac{1}{2}} U\|_2 | \mathscr{A}_T] \leq \mathbb{E}[\|C_{x,T}^{-\frac{1}{2}} - C_x^{-\frac{1}{2}}\|_2 \|U\|_2 | \mathscr{A}_T] = \mathbb{E}[\|C_{x,T}^{-\frac{1}{2}} - C_x^{-\frac{1}{2}}\|_2 | \mathscr{A}_T]$$

$$\mathbb{E}[\|\tilde{V} - \widehat{V}\|_2 | \mathscr{A}_T] = \mathbb{E}[\|C_{y,T}^{-\frac{1}{2}} V - C_y^{-\frac{1}{2}} V\|_2 | \mathscr{A}_T] \leq \mathbb{E}[\|C_{y,T}^{-\frac{1}{2}} - C_y^{-\frac{1}{2}}\|_2 \|V\|_2 | \mathscr{A}_T] = \mathbb{E}[\|C_{y,T}^{-\frac{1}{2}} - C_y^{-\frac{1}{2}}\|_2 | \mathscr{A}_T]$$

The proof simply follows from the following equations:

$$\begin{aligned}
\mathbb{E}\left[\left\|C_x^{-\frac{1}{2}} - C_{x,T}^{-\frac{1}{2}}\right\|_2 \Big| \mathscr{A}_T\right] &= \mathbb{E}\left[\left\|C_x^{-\frac{1}{2}} \left(C_{x,T}^{\frac{1}{2}} - C_x^{\frac{1}{2}}\right) C_{x,T}^{-\frac{1}{2}}\right\|_2 \Big| \mathscr{A}_T\right] \\
&\leq \mathbb{E}\left[\left\|C_x^{-\frac{1}{2}}\right\|_2 \left\|C_{x,T}^{-\frac{1}{2}}\right\|_2 \left\|C_{x,T}^{\frac{1}{2}} - C_x^{\frac{1}{2}}\right\|_2 \Big| \mathscr{A}_T\right] \\
&\leq \frac{\sqrt{2}}{r_x} \mathbb{E}\left[\left\|C_{x,T}^{\frac{1}{2}} - C_x^{\frac{1}{2}}\right\|_2 \Big| \mathscr{A}_T\right] \\
&\leq \frac{\sqrt{2}}{(1 + \sqrt{2}/2) r_x^{3/2}} \mathbb{E}\left[\|C_{x,T} - C_x\|_2 | \mathscr{A}_T\right] \leq \frac{1}{r_x^{3/2}} \mathbb{E}\left[\|C_{x,T} - C_x\|_2 | \mathscr{A}_T\right] \\
&\quad\quad\quad\quad\quad\quad\quad\quad\quad\quad\quad\quad\quad\quad\quad\quad\quad \text{(by Lemma A.4)}
\end{aligned}$$

and the fact that by Lemma A.5 $\mathbb{E}\left[\|C_{x,T} - C_x\|_2 | \mathscr{A}_T\right] \leq \sqrt{\frac{2B^2}{T} \log(d_x)} + \frac{2B}{3T} \log(d_x)$. $\square$

*Proof of Theorem 2.4.* First note that

$$\text{Tr}(U_*^\top C_{xy} V_* - \tilde{U}^\top C_{xy} \tilde{V}) = \text{Tr}(U_*^\top C_{xy} V_* - \widehat{U}^\top C_{xy} \widehat{V}) + \text{Tr}(\widehat{U}^\top C_{xy} \widehat{V} - \tilde{U}^\top C_{xy} \tilde{V})$$

Moreover, we have that $\text{Tr}(\widehat{U}^\top C_{xy} \widehat{V} - \tilde{U}^\top C_{xy} \tilde{V}) \leq 2k\|\widehat{U}^\top C_{xy} \widehat{V} - \tilde{U}^\top C_{xy} \tilde{V}\|_2$. We bound the right hand side using the following equations

$$
\begin{aligned}
\mathbb{E}[\|\widehat{U}^\top C_{xy}\widehat{V} - \tilde{U}^\top C_{xy}\tilde{V}\|_2 | \mathscr{A}_T] &= \mathbb{E}[\|\widehat{U}^\top C_{xy}\widehat{V} - \tilde{U}^\top C_{xy}\widehat{V} + \tilde{U}^\top C_{xy}\widehat{V} - \tilde{U}^\top C_{xy}\tilde{V}\|_2 | \mathscr{A}_T] \\
&\leq \mathbb{E}[\|(\widehat{U} - \tilde{U})^\top C_{xy}\widehat{V}\|_2 + \|\tilde{U}^\top C_{xy}(\widehat{V} - \tilde{V})\|_2 | \mathscr{A}_T] \\
&\qquad\qquad\qquad\qquad\qquad\qquad\qquad \text{(triangle inequality)} \\
&\leq \mathbb{E}[\|\widehat{U} - \tilde{U}\|_2 \|C_{xy}\|_2 \|\widehat{V}\|_2 + \|\tilde{U}\|_2 \|C_{xy}\|_2 \|\widehat{V} - \tilde{V}\|_2 | \mathscr{A}_T] \\
&\qquad\qquad\qquad \text{(sub-multiplicativity of the operator norm)} \\
&\leq B\mathbb{E}[\|\widehat{U} - \tilde{U}\|_2 \|C_y^{-\frac{1}{2}} V\|_2 + B\|C_{x,t}^{-\frac{1}{2}} U\|_2 \|\widehat{V} - \tilde{V}\|_2 | \mathscr{A}_T] \\
&\leq \mathbb{E}[\|\widehat{U} - \tilde{U}\|_2 | \mathscr{A}_T]\frac{B}{\sqrt{r_y}} + \frac{B}{\sqrt{r_x}}\mathbb{E}[\|\widehat{V} - \tilde{V}\|_2 | \mathscr{A}_T] \\
&\leq \frac{B}{2r_x^2}\left(\sqrt{\frac{2B^2}{T}\log(d_x)} + \frac{2B}{3T}\log(d_x)\right) \quad \text{(by Lemma A.8)} \\
&\quad + \frac{B}{2r_y^2}\left(\sqrt{\frac{2B^2}{T}\log(d_y)} + \frac{2B}{3T}\log(d_y)\right) \\
&\leq \frac{B}{r^2}\left(\sqrt{\frac{2B^2}{T}\log(d)} + \frac{2B}{3T}\log(d)\right)
\end{aligned}
$$

which completes the first part of the proof. For the second part of the proof it holds:

$$
\begin{aligned}
\left\|\tilde{U}^\top C_x \tilde{U} - I\right\|_2 &= \left\|\tilde{U}^\top C_x \tilde{U} - \widehat{U}^\top C_x \widehat{U}\right\|_2 \\
&\leq \left\|\tilde{U}^\top C_x \tilde{U} - \widehat{U}^\top C_x \tilde{U}\right\|_2 + \left\|\widehat{U}^\top C_x \tilde{U} - \widehat{U}^\top C_x \widehat{U}\right\|_2 \\
&\leq \left(\left\|C_x \tilde{U}\right\|_2 + \left\|\widehat{U}^\top C_x\right\|_2\right)\left\|\tilde{U} - \widehat{U}\right\|_2 \\
&\leq B\left(\left\|C_{x,t}^{-1/2}\right\|_2 \|U\|_2 + \left\|C_x^{-1/2}\right\|_2 \|U\|_2\right)\left\|\tilde{U} - \widehat{U}\right\|_2 \\
&\leq \frac{2B}{\sqrt{r_x}}\left\|\tilde{U} - \widehat{U}\right\|_2.
\end{aligned}
$$

Applying lemma A.8 allows us to bound $\mathbb{E}\left[\left\|\tilde{U}^\top C_x \tilde{U} - I\right\|_2 | \mathscr{A}_T\right]$. Using Law of Total Expectation we get:

$$
\begin{aligned}
\mathbb{E}\left[\left\|\tilde{U}^\top C_x \tilde{U} - I\right\|_2\right] &= \mathbb{E}\left[\left\|\tilde{U}^\top C_x \tilde{U} - I\right\|_2 | \mathscr{A}_T\right](1 - \delta) + \mathbb{E}\left[\left\|\tilde{U}^\top C_x \tilde{U} - I\right\|_2 | \bar{\mathscr{A}}_T\right]\delta \\
&\leq \mathbb{E}\left[\left\|\tilde{U}^\top C_x \tilde{U} - I\right\|_2 | \mathscr{A}_T\right] + \delta\mathbb{E}\left[\left\|\tilde{U}^\top C_x \tilde{U} - I\right\|_2 | \bar{\mathscr{A}}_T\right] \\
&\leq \frac{B}{r_x^2}\left(\sqrt{\frac{2B^2}{T}\log(d_x)} + \frac{2B}{3T}\log(d_x)\right) + \delta(B + 1).
\end{aligned}
$$

Setting $\delta = \frac{1}{T}$ finishes the proof of the second part. The third inequality of the theorem follows similarly. $\qquad\square$

## B   Matrix Exponentiated Gradient for CCA

To make analysis easier, in this section we decide to analyze Algorithm 2 for solving a rescaled version of problem 12. In particular, we rescale the constraints in 12 so that the feasible set becomes the set of density matrices, $\{M : \text{Tr}(M) = 1 \text{ and } 0 \preceq M \preceq \frac{1}{k}I\}$. The results in section 3 are recovered from the proofs presented here by rescaling all bounds by a factor of $k$. We denote the error in the gradient at time $t$ by $\bar{E}_t = C_t - \tilde{C}_t$ and $E_t = g_t - \partial_t$. We will need the following lemmas from [22].

**Lemma B.1** (Golden-Thompson inequality [9]). *For arbitrary symmetric matrices* A *and* B, *it holds:*

$$\mathrm{Tr}\left(\exp\left(\mathrm{A}+\mathrm{B}\right)\right) \le \mathrm{Tr}\left(\exp\left(A\right)\exp\left(B\right)\right).$$

**Lemma B.2.** *For any PSD matrix* A *and symmetric* B, C, $\mathrm{B} \preceq \mathrm{C}$ *implies* $\mathrm{Tr}\left(\mathrm{AB}\right) \le \mathrm{Tr}\left(\mathrm{AC}\right)$.

**Lemma B.3.** *For any symmetric* A *such that* $0 \preceq \mathrm{A} \preceq \mathrm{I}$ *and any* $\rho_1, \rho_2 \in \mathbb{R}$ *the following holds*

$$\exp\left(\mathrm{A}\rho_1 + \left(\mathrm{I}-\mathrm{A}\right)\rho_2\right) \preceq \mathrm{A}\exp\left(\rho_1\right) + \left(\mathrm{I}-\mathrm{A}\right)\exp\left(\rho_2\right).$$

We also need the following lemma.

**Lemma B.4.** *For* $x = 1 + \sqrt{\frac{R}{L}}$ *the following holds*

$$\frac{-R + \log\left(x\right)L}{x-1} \ge L - 2\sqrt{RL}.$$

*Proof.* This is a simple consequence of the fact that $\log\left(x\right) \ge \left(x-1\right) - \left(x-1\right)^2$ for $x \ge 1$. $\square$

---

**Algorithm 2** Matrix Exponentiated Gradient for CCA (MEG-CCA)

---

**Input:** Training data $\{(\mathrm{x}_t, \mathrm{y}_t)\}_{t=1}^T$, step size $\eta$, auxiliary training data $\{(\mathrm{x}_i', \mathrm{y}_i')\}_{t=i}^\tau$
**Output:** $\tilde{\mathrm{M}}$
    Initialize: $\mathrm{M}_0 \leftarrow \frac{1}{d}\mathrm{I}$, $\mathrm{C}_{x,0} \leftarrow \frac{1}{\tau}\sum_{i=1}^\tau \mathrm{x}_i'\mathrm{x}_i'^\top$, $\mathrm{C}_{y,0} \leftarrow \frac{1}{\tau}\sum_{i=1}^\tau \mathrm{y}_i'\mathrm{y}_i'^\top$
    **for** $t = 1$ to $T$ **do**
        $\mathrm{C}_{x,t} \leftarrow \frac{t+\tau-1}{t+\tau}\mathrm{C}_{x,t-1} + \frac{1}{t+\tau}\mathrm{x}_t\mathrm{x}_t^\top$, $\mathrm{W}_{x,t} \leftarrow \mathrm{C}_{x,t}^{-\frac{1}{2}}$
        $\mathrm{C}_{y,t} \leftarrow \frac{t+\tau-1}{t+\tau}\mathrm{C}_{y,t-1} + \frac{1}{t+\tau}\mathrm{y}_t\mathrm{y}_t^\top$, $\mathrm{W}_{y,t} \leftarrow \mathrm{C}_{y,t}^{-\frac{1}{2}}$
        $\tilde{\mathrm{C}}_t \leftarrow \begin{pmatrix} 0 & \partial_t \\ \partial_t^\top & 0 \end{pmatrix} = \frac{1}{2}\begin{pmatrix} \mathrm{W}_{x,t}\mathrm{x}_t \\ \mathrm{W}_{y,t}\mathrm{y}_t \end{pmatrix}\begin{pmatrix} \mathrm{W}_{x,t}\mathrm{x}_t \\ \mathrm{W}_{y,t}\mathrm{y}_t \end{pmatrix}^\top - \frac{1}{2}\begin{pmatrix} \mathrm{W}_{x,t}\mathrm{x}_t \\ -\mathrm{W}_{y,t}\mathrm{y}_t \end{pmatrix}\begin{pmatrix} \mathrm{W}_{x,t}\mathrm{x}_t \\ -\mathrm{W}_{y,t}\mathrm{y}_t \end{pmatrix}^\top$
        $\widehat{\mathrm{M}}_t \leftarrow \frac{\exp\left(\log(\mathrm{M}_{t-1})+\eta\tilde{\mathrm{C}}_t\right)}{\mathrm{Tr}\left(\exp\left(\log(\mathrm{M}_{t-1})+\eta\tilde{\mathrm{C}}_t\right)\right)}$
        $\mathrm{M}_t \leftarrow \mathscr{P}\left(\widehat{\mathrm{M}}_t\right)$         % projection is given by algorithm 4 in [27]
    **end for**
    $\bar{\mathrm{M}} = \frac{1}{T}\sum_{t=1}^T \mathrm{M}_{t-1}$
    $\tilde{\mathrm{M}} = \texttt{rounding}\left(\bar{\mathrm{M}}\right)$

---

**Lemma B.5.** *Conditioned on the event* $\mathscr{A}_T$ *occurring, after* $T$ *iterations of Algorithm 2 with a step size* $\eta = \frac{1}{G}\log\left(1 + \sqrt{\frac{\log(d)}{GT}}\right)$, *where* $G = \frac{2B}{\sqrt{r_x r_y}}$ *and* $\mathrm{M}_0 = \frac{1}{d}\mathrm{I}$ *we have that,*

$$\sum_{t=1}^T \mathrm{Tr}\left(\mathrm{M}_*\tilde{\mathrm{C}}_t\right) - \sum_{t=1}^T \mathrm{Tr}\left(\mathrm{M}_{t-1}\tilde{\mathrm{C}}_t\right) \le 2\sqrt{G^2 T \log\left(d\right)}, \tag{18}$$

*where* $\mathrm{M}_*$ *is an optimum of Problem (11).*

*Proof of Lemma B.5.* The proof closely follows proof of Lemma 3.1 in [22], however, we provide it for completeness. Lemma 2.1 implies that $\mathrm{W}_{x,t} \preceq \sqrt{\frac{2}{r_x}}$ and $\mathrm{W}_{y,t} \preceq \sqrt{\frac{2}{r_y}}$ with probability $1-\delta$. Let $F(\mathrm{W}) = \mathrm{Tr}\left(\mathrm{W}\log\left(\mathrm{W}\right) - \mathrm{W}\right)$ be the von Neumann entropy and denote by $\Delta(\mathrm{A},\mathrm{B})$, the von Neumann divergence induced by $F(\cdot)$. More precisely, $\Delta(\mathrm{A},\mathrm{B}) = \mathrm{Tr}\left(\mathrm{A}\log\left(\mathrm{A}\right) - \mathrm{A}\log\left(\mathrm{B}\right) - \mathrm{A} + \mathrm{B}\right)$. First we note that the update step (13) (after substituting $\mathrm{C}_t$ with $\tilde{\mathrm{C}}_t$) is invariant under perturbing the $\tilde{\mathrm{C}}_t$'s by a multiple of the identity [25], so we can assume that each $\tilde{\mathrm{C}}_t \succeq 0$. Since $\max\left(\|x_t\|^2, \|y_t\|^2\right) \le B$, $\mathrm{W}_{x,t} \preceq \sqrt{\frac{2}{r_x}}\mathrm{I}$ and $\mathrm{W}_{y,t} \preceq \sqrt{\frac{2}{r_y}}\mathrm{I}$, we see that $G = \frac{2B}{\sqrt{r_x r_y}}$ is such that $\tilde{\mathrm{C}}_t - \lambda_{\min}\left(\tilde{\mathrm{C}}_t\right)\mathrm{I} \preceq G\mathrm{I}$. Also it holds that $\mathrm{Tr}\left(\mathrm{M}^*\tilde{\mathrm{C}}_t\right) \le \|\mathrm{M}^*\|_2 \left\|\tilde{\mathrm{C}}_t\right\|_* \le 2\left\|\tilde{\mathrm{C}}_t\right\|_2 \le G$,

where the second to last inequality holds because $\tilde{C}_t$ is a rank-2 matrix for all $t$. We begin by considering the difference $\Delta\left(M, M_{t-1}\right) - \Delta\left(M, \widehat{M}_t\right)$ for any $M$ in the feasible set of (12).

$$\Delta\left(M, M_{t-1}\right) - \Delta\left(M, \widehat{M}_t\right) = \operatorname{Tr}\left(M(\log\left(M\right) - \log\left(M_{t-1}\right))\right) - \operatorname{Tr}\left(M\left(\log\left(M\right) - \log\left(\widehat{M}_t\right)\right)\right)$$

$$= -\operatorname{Tr}\left(M\left(\log\left(M_{t-1}\right) - \log\left(\frac{\exp\left(\log\left(M_{t-1}\right) + \eta\tilde{C}_t\right)}{\operatorname{Tr}\left(\exp\left(\log\left(M_{t-1}\right) + \eta\tilde{C}_t\right)\right)}\right)\right)\right)$$

$$= \eta\operatorname{Tr}\left(M\tilde{C}_t\right) - \log\left(\operatorname{Tr}\left(\exp\left(\log\left(M_{t-1}\right) + \eta\tilde{C}_t\right)\right)\right),$$

where the first equality holds by the fact $\operatorname{Tr}\left(M\right) = \operatorname{Tr}\left(M_{t-1}\right)$ and the second inequality holds by expanding $\widehat{M}_t$, according to (13). We now bound $\log\left(\operatorname{Tr}\left(\exp\left(\log\left(M_{t-1}\right) + \eta\tilde{C}_t\right)\right)\right)$. By Golden-Thompson's inequality B.1, we have

$$\operatorname{Tr}\left(\exp\left(\log\left(M_{t-1}\right) + \eta\tilde{C}_t\right)\right) \leq \operatorname{Tr}\left(M_{t-1}\exp\left(\eta\tilde{C}_t\right)\right).$$

Next, since $0 \preceq \frac{\tilde{C}_t}{G} \preceq I$, we use Lemma B.3 on $\exp\left(\eta\tilde{C}_t\right)$ with $\rho_0 = 0$ and $\rho_1 = G\eta$ to get $\exp\left(\eta\tilde{C}_t\right) \preceq \frac{\tilde{C}_t}{G}(\exp\left(G\eta\right) - 1) + I$. By Lemma B.2, we now have

$$\operatorname{Tr}\left(M_{t-1}\exp\left(\eta\tilde{C}_t\right)\right) \leq \operatorname{Tr}\left(M_{t-1} + M_{t-1}\frac{\tilde{C}_t}{G}(\exp\left(G\eta\right) - 1)\right),$$

which implies

$$\log\left(\operatorname{Tr}\left(M_{t-1}\exp\left(\eta\tilde{C}_t\right)\right)\right) \leq \log\left(1 + \frac{\operatorname{Tr}\left(M_{t-1}\tilde{C}_t\right)}{G}(\exp\left(G\eta\right) - 1)\right) \leq \frac{\operatorname{Tr}\left(M_{t-1}\tilde{C}_t\right)}{G}(\exp\left(G\eta\right) - 1),$$

where last inequality holds since $\log\left(1 + x\right) \leq x$. Thus

$$\Delta\left(M, M_{t-1}\right) - \Delta\left(M, \widehat{M}_t\right) \geq \eta\operatorname{Tr}\left(M\tilde{C}_t\right) - (\exp\left(G\eta\right) - 1)\frac{\operatorname{Tr}\left(M_{t-1}\tilde{C}_t\right)}{G}.$$

Equivalently,

$$\operatorname{Tr}\left(M_{t-1}\tilde{C}_t\right) \geq G\frac{\Delta\left(M, \widehat{M}_t\right) - \Delta\left(M, M_{t-1}\right) + \eta\operatorname{Tr}\left(M\tilde{C}_t\right)}{\exp\left(G\eta\right) - 1}.$$

By Generalized Pythagorean Theorem

$$\operatorname{Tr}\left(M_{t-1}\tilde{C}_t\right) \geq G\frac{\Delta\left(M, M_t\right) - \Delta\left(M, M_{t-1}\right) + \eta\operatorname{Tr}\left(M\tilde{C}_t\right)}{\exp\left(G\eta\right) - 1}.$$

Summing from $t = 1$ to $T$ and using the fact the Bregman divergence is positive we have

$$\sum_{t=1}^{T}\operatorname{Tr}\left(M_{t-1}\tilde{C}_t\right) \geq G\frac{-\Delta\left(M, M_0\right) + \eta\sum_{t=1}^{T}\operatorname{Tr}\left(M_*\tilde{C}_t\right)}{\exp\left(G\eta\right) - 1}.$$

To complete the proof notice that $\Delta\left(M, M_0\right) \leq \log\left(d\right)$ and apply lemma B.4 with $\eta = \frac{1}{G}\log\left(1 + \sqrt{\frac{G\log(d)}{GT}}\right)$. $\qquad\square$

**Lemma B.6.** *Assume that the event $\mathscr{A}_t$ occurs and that $E_t$ has no repeated singular values. It holds that*

$$-\frac{\kappa}{\sqrt{t}}I \preceq \mathbb{E}_{x_t, y_t}\left[\bar{E}_t | \mathscr{A}_t\right] \preceq \frac{\kappa}{\sqrt{t}}I.$$

*Proof.* By the properties of self-adjoint dilation, we have $\mathbb{E}_{x_t, y_t}\left[\|E_t\|_2 | \mathscr{A}_t\right] = \mathbb{E}_{x_t, y_t}\left[\|\bar{E}_t\|_2 | \mathscr{A}_t\right]$. By Jensen's inequality and Lemma 2.2 we have $\left\|\mathbb{E}_{x_t, y_t}\left[\bar{E}_t | \mathscr{A}_t\right]\right\|_2 \leq \mathbb{E}_{x_t, y_t}\left[\|\bar{E}_t\|_2 | \mathscr{A}_t\right] = \mathbb{E}_{x_t, y_t}\left[\|E_t\|_2 | \mathscr{A}_t\right] \leq \frac{\kappa}{\sqrt{t}}$ and thus $-\frac{\kappa}{\sqrt{t}}I \preceq \mathbb{E}_{x_t, y_t}\left[\bar{E}_t | \mathscr{A}_t\right] \preceq \frac{\kappa}{\sqrt{t}}I$. $\qquad\square$

*Proof of Lemma 3.1.* Since $M_{t-1}$ is independent of $(x_t, y_t)$ we have $\mathbb{E}_{x_t,y_t}\left[\text{Tr}\left(M_{t-1}\bar{E}_t\right)|\mathscr{A}_t\right]$ $= \text{Tr}\left(M_{t-1}\mathbb{E}_{x_t,y_t}\left[\bar{E}_t|\mathscr{A}_t\right]\right)$. From Lemma B.6, we know that $\mathbb{E}_{x_t,y_t}\left[\bar{E}_t|\mathscr{A}_t\right] \preceq \frac{\kappa}{\sqrt{t}}I$ and since $M_{t-1} \succeq 0$, Lemma B.2 implies $\text{Tr}\left(M_{t-1}\mathbb{E}_{x_t,y_t}\left[\bar{E}_t|\mathscr{A}_t\right]\right) \leq \frac{\kappa}{\sqrt{t}}\text{Tr}\left(M_{t-1}\right) = \frac{\kappa}{\sqrt{t}}$. Similarly using that $-\frac{\kappa}{\sqrt{t}}I \preceq \mathbb{E}_{x_t,y_t}\left[\bar{E}_t|\mathscr{A}_t\right]$, we have $\mathbb{E}_{x_t,y_t}\left[\text{Tr}\left(M_*\bar{E}_t\right)|\mathscr{A}_t\right] \geq -\frac{\kappa}{\sqrt{t}}$ and the result follows. □

*Proof of Theorem 3.2.* By Lemma B.5, we have

$$\sum_{t=1}^{T}\text{Tr}\left(M_*\left(C_t - \bar{E}_t\right)\right) - \sum_{t=1}^{T}\text{Tr}\left(M_{t-1}\left(C_t - \bar{E}_t\right)\right) \leq 2\sqrt{G^2 T \log\left(d\right)}. \tag{19}$$

Let $\mathbb{E}_\tau\left[\cdot\right]$ denote the expectation w.r.t. $(x_t, y_t)_{t=1}^\tau$. We now compute the expectations of the two terms on the left hand side of (19)

$$\sum_{t=1}^{T}\mathbb{E}\left[\text{Tr}\left(M_*\left(C_t - \bar{E}_t\right)\right)|\mathscr{A}_t\right] = T\text{Tr}\left(M_*C\right) - \sum_{t=1}^{T}\mathbb{E}\left[\text{Tr}\left(M_*\bar{E}_t\right)|\mathscr{A}_t\right] \tag{20}$$

The second term expands as follows

$$\sum_{t=1}^{T}\mathbb{E}\left[\text{Tr}\left(M_{t-1}\left(C_t - \bar{E}_t\right)\right)|\mathscr{A}_t\right] = \sum_{t=1}^{T}\text{Tr}\left(\mathbb{E}_t\left[M_{t-1}\left(C_t - \bar{E}_t\right)|\mathscr{A}_t\right]\right)$$

$$= \sum_{t=1}^{T}\text{Tr}\left(\mathbb{E}_{t-1}\left[\mathbb{E}_t\left[M_{t-1}\left(C_t - \bar{E}_t\right)|(x_i, y_i)_{i=1}^{t-1}, \mathscr{A}_t\right]\right]\right)$$

$$= \sum_{t=1}^{T}\text{Tr}\left(\mathbb{E}_{t-1}\left[M_{t-1}|\mathscr{A}_t\right]\mathbb{E}_t\left[\left(C_t - \bar{E}_t\right)|\mathscr{A}_t\right]\right)$$

$$= \sum_{t=1}^{T}\mathbb{E}\left[\text{Tr}\left(M_{t-1}C\right)|\mathscr{A}_t\right] - \sum_{t=1}^{T}\text{Tr}\left(\mathbb{E}_{t-1}\left[M_{t-1}|\mathscr{A}_t\right]\mathbb{E}_t\left[\bar{E}_t|\mathscr{A}_t\right]\right)$$

$$= \sum_{t=1}^{T}\mathbb{E}\left[\text{Tr}\left(M_{t-1}C\right)|\mathscr{A}_t\right] - \sum_{t=1}^{T}\mathbb{E}_{t-1}\left[\mathbb{E}_{x_t,y_t}\left[\text{Tr}\left(M_{t-1}\bar{E}_t\right)|\mathscr{A}_t\right]\right],$$

$$\tag{21}$$

where the second equality holds by smoothing property of expectation and the third equality holds because $M_{t-1}$ is conditionally independent of $C_t$ and $\bar{E}_t$. Putting together (19), (20), (21) we have

$$T\text{Tr}\left(M_*C\right) - \sum_{t=1}^{T}\mathbb{E}\left[\text{Tr}\left(M_{t-1}C\right)|\mathscr{A}_t\right]$$

$$\leq 2\sqrt{G^2 T \log\left(d\right)} + \sum_{t=1}^{T}\left[\mathbb{E}_{t-1}\left[\mathbb{E}_{x_t,y_t}\left[\text{Tr}\left(M_{t-1}\bar{E}_t\right)|\mathscr{A}_t\right]\right] - \mathbb{E}\left[\text{Tr}\left(M_*\bar{E}_t\right)|\mathscr{A}_t\right]\right]$$

$$\tag{22}$$

$$= 2\sqrt{G^2 T \log\left(d\right)} + \sum_{t=1}^{T}\mathbb{E}_{t-1}\left[\mathbb{E}_{x_t,y_t}\left[\text{Tr}\left(M_{t-1}\bar{E}_t\right) - \text{Tr}\left(M_*\bar{E}_t\right)|\mathscr{A}_t\right]\right]$$

$$\leq 2\sqrt{G^2 T \log\left(d\right)} + \sum_{t=1}^{T}\frac{\kappa}{\sqrt{t}} \leq 2\sqrt{G^2 T \log\left(d\right)} + 2\sqrt{T}\kappa$$

where the second inequality follows from Lemma 3.1 and the last inequality follows from Lemma A.6. Next we bound:

$$T\text{Tr}\left(M_*C\right) - \sum_{t=1}^{T}\mathbb{E}\left[\text{Tr}\left(M_{t-1}C\right)\right] = T\text{Tr}\left(M_*C\right) - \sum_{t=1}^{T}\mathbb{E}\left[\text{Tr}\left(M_{t-1}C\right)|\mathscr{A}_t\right]\left(1 - \delta\right) - \sum_{t=1}^{T}\mathbb{E}\left[\text{Tr}\left(M_{t-1}C\right)|\bar{\mathscr{A}}_t\right]\delta$$

$$\leq T\text{Tr}\left(M_*C\right) - \sum_{t=1}^{T}\mathbb{E}\left[\text{Tr}\left(M_{t-1}C\right)|\mathscr{A}_t\right] - \sum_{t=1}^{T}\mathbb{E}\left[\text{Tr}\left(M_{t-1}C\right)|\bar{\mathscr{A}}_t\right]\delta$$

$$\leq 2\sqrt{G^2 T \log\left(d\right)} + 2\sqrt{T}\kappa.$$

To finish the proof we only need to divide both sides by $T$. □