[Reviews · NeurIPS 2017]

Reviewer 1



The authors propose a stochastic approximation of CCA which sound reasonnalbe and technically correct. They start with a description of CCA and regularized CCA and conclude with a least squares formulation of rCCA. They proposed a convex relaxation of rCCA easier to extend to the stochastic context. The correponding algorithm (MSG-CCA) is then derived and relies on the estimation of the gradient of the objective function. A simple formulation for the approximation of the gradient is proposed. Then two bounds are given. The first one bounds the size of the expected noise in the estimate of the inexact gradient The second one bounds the suboptimality gap in the CCA objective between the true optimal rank-k subspace and the rank-k subspace returned by MSG-CCA after a number of iterations Motivated by its efficiency for stochastic PCA and stochastic PLS, a formulation of the algorithm based on Matrix Exponentiated Gradient (MEG-SCCA) is proposed. These two algorithm are compared in the Experiments section on the real dataset Mediamill (a two-view dataset consisting of 10000 videos and text annotations) and compared to competitive approaches in term of CCA objective, CPU time and number of iterations. We mention that for MSG-CCA and MEG-CCA, the regularization parameters and the step size that has to be tuned are fixed with no justification.

Reviewer 2



I acknowledge having read the rebuttal. I thank for the clarifications regarding the experiment, but I did not find the answer particularly convincing -- I simply do not understand why you would use a medium-scale data set that is too small to show the difference, and the rebuttal did not really clarify this. Since you did not even hint at what kind of results the omitted artificial data experiments would provide, we cannot evaluate whether they would strengthen the paper or not. The paper proposes a stochastic approximation algorithm for CCA, and provide theoretical analysis of the algorithm in terms of \epsilon-suboptimality. It is empirically demonstrated to outperform earlier stochastic approximation algorithms that optimize the empirical estimate. This is overall a solid theoretical work. While we have seen a few stochastic algorithms for CCA in recent years, the previous work has completely ignored the generalization ability that is in focus here. The proposed solution and its theoretical analysis seem to be sound, and I can foresee other potential applications for the matrix stochastic gradient approach. The practical impact of the actual computational method is, however, somewhat limited. Figure 1 illustrates a CPU gain of roughly one order of magnitude over CCA-Lin, but given that the experiment is on a single medium-scale data set we cannot draw very strong conclusions -- there is no evidence that some practical problem would be solved by this method while being infeasible for the existing ones. The poor empirical section clearly degrades the value of the paper. Issues: 1. Introduction is missing some references; even though this is a theoretical paper it would make sense to cite some works when motivating the multi-view setup -- combining "tweets + friends network" sounds like it refers to a particular work, yet it is not cited. Also, "It can be shown that CCA is non-learnable" would need a justification. 2. The experiments are too hurried. You never explain what "batch" refers to (batch gradient descent for Eq.(4)?). More importantly, you do not comment in any way the fact that especially for k=4 it is actually clearly the fastest algorithm in run-time and competitive for other k as well. This partly defeats the purpose of the paper, but I wonder whether this is simply because of poor choice of data. Perhaps you could consider running a somewhat larger (say, n=10^6 or more) example instead to demonstrate the practical importance? Some comment on limiting to only very small values of k would also be crucial, since all examples used to motivate the work require values of k in the order of tens or hundreds. Without evidence on efficient computation for large-scale problems the Introduction that motivates the work in part by "scaling to very large datasets" is misleading, leaving online learning as the only advantage. 3. Section 1.1 has Frobenius and nuclear norm in the wrong order.

Reviewer 3



I read the rebuttal and thank the authors for the clarifications about differences between the proposed method and previous work in [3,4]. I agreed with Reviewer 2's concerns about the limits of the evaluation section. Giving experiments on only one medium scale data set, it is insufficient to show the potential computational advantages of the proposed method (scale easily to very large datasets) compared to the alternative methods. Summary The paper proposes two novel first-order stochastic optimization algorithms to solve the online CCA problem. Inspired from previous work, the proposed algorithms are instances of noisy matrix stochastic / exponentiated gradient (MSG / MEG): at each iteration, for a new sample (x_t , y_t), the algorithms work by first updating the empirical estimates of the whitening transformations which define the inexact gradient partial_t, followed by a projection step over the constrain set of the problem to get a rank-k projection matrix (M_t). The theoretical analysis and experimental results show the superiority of the algorithms compared to state-of-the-art methods for CCA. Qualitative Assessment Significance - Justification: The main contributions of the paper are (1) the derivation of a new noisy stochastic mirror descent method to solve a convex relaxation for the online version of CCA and (2) the provision of generalization bounds (or convergence guarantees) for the proposed algorithms. These contributions appear to be incremental advances considered that similar solutions have been proposed previously for PCA [3] and PLS [4]. Detailed comments: The paper on the whole is well written, the algorithms are explained clearly, the theorems for the algorithms are solid and sound, and the performance of the algorithms outperforms that of state-of-the-art methods for CCA. Some questions/comments: What differentiates your algorithm from that of the previous work in references [3, 4]? Except solving different problems (CCA vs PCA and PLS), is your algorithm fundamentally different in some ways (e.g. add whitening transformations to alleviate the gradient updates for CCA problems or provide convergence analysis)? The reviewer read references [3, 4] and noticed that the algorithms in your papers are similar to those proposed in [3, 4] in terms of similar convex relaxation and stochastic optimization algorithms. The reviewer would have expected the authors to provide a comparison/summary of the differences / similarities between previous work [3, 4, 21] and current paper in the related work. This would also highlight explicitly the main contributions of the paper. There are a few typos: Line 90: build -> built; study -> studied Line 91: in the batch setting -> in a batch setting Line 96: pose -> posed; consider -> considered Line 101-102: rewrite this sentence “none of the papers above give generalization bounds and focus is entirely on analyzing the quality of the solution to that of the empirical minimizer”. I guess you intend to express the following: “all the papers mentioned above focus entirely / mainly on analyzing the quality of the solution to that of the empirical minimizer, but provide no generalization bounds” remove line 150